# Piecewise deterministic generative models

**Andrea Bertazzi**[1,*]**, Dario Shariatian**[2]
**Umut Simsekli**[2]**, Eric Moulines**[1,3]**, Alain Durmus**[1]
[1] École Polytechnique, Institut Polytechnique de Paris
[2] INRIA, CNRS, Ecole Normale Supérieure, PSL Research University
[3] MBZUAI
[*] `andrea.bertazzi@polytechnique.edu`

## Abstract

We introduce a novel class of generative models based on piecewise deterministic Markov processes (PDMPs), a family of non-diffusive stochastic processes consisting of deterministic motion and random jumps at random times. Similarly to diffusions, such Markov processes admit time reversals that turn out to be PDMPs as well. We apply this observation to three PDMPs considered in the literature: the Zig-Zag process, Bouncy Particle Sampler, and Randomised Hamiltonian Monte Carlo. For these three particular instances, we show that the jump rates and kernels of the corresponding time reversals admit explicit expressions depending on some conditional densities of the PDMP under consideration before and after a jump. Based on these results, we propose efficient training procedures to learn these characteristics and consider methods to approximately simulate the reverse process. Finally, we provide bounds in the total variation distance between the data distribution and the resulting distribution of our model in the case where the base distribution is the standard $d$-dimensional Gaussian distribution. Promising numerical simulations support further investigations into this class of models.

## 1 Introduction

Diffusion-based generative models [Ho et al., 2020, Song et al., 2021] have recently achieved state-of-the-art performance in various fields of application [Dhariwal and Nichol, 2021, Croitoru et al., 2023, Jeong et al., 2021, Kong et al., 2021]. In their continuous time interpretation [Song et al., 2021], these models leverage the idea that a diffusion process can bridge the data distribution $\mu_\star$ to a base distribution $\pi$, and its time reversal can transform samples from $\pi$ into synthetic data from $\mu_\star$. Anderson [1982] showed that the time reversal of a diffusion process, i.e., the *backward* process, is itself a diffusion with explicit drift and covariance functions that are related to the score functions of the time-marginal densities of the original, *forward* diffusion. Consequently, the key element of these generative models is learning these score functions using techniques such as (denoising) score-matching [Hyvärinen, 2005, Vincent, 2011].

In this work we propose a new family of generative models which use piecewise deterministic Markov processes (PDMPs) as noising processes instead of diffusions. PDMPs were introduced around forty years ago [Davis, 1984, 1993] and since then have been successfully applied in various fields, including communication networks [Dumas et al., 2002], biology [Berg and Brown, 1972, Cloez, Bertrand et al., 2017], risk theory [Embrechts and Schmidli, 1994], and the reliability of complex systems [Zhang et al., 2008]. More recently, PDMPs have been intensively studied in the context of Monte Carlo algorithms [Fearnhead et al., 2018] as alternatives to Langevin diffusion-based methods and Metropolis-Hastings mechanisms. This renewed interest in PDMPs has led to the development of novel processes, such as the Zig-Zag process (ZZP) [Bierkens et al., 2019a], the Bouncy Particle Sampler (BPS) [Bouchard-Côté et al., 2018], and the Randomised Hamiltonian

38th Conference on Neural Information Processing Systems (NeurIPS 2024).

Monte Carlo (RHMC) [Bou-Rabee and Sanz-Serna, 2017]. PDMPs offer several advantages compared to Langevin-based methods, such as better scalability and reduced computational complexity in high-dimensional settings [Bierkens et al., 2019a]. In the context of generative modelling PDMPs offer several potential advantages over diffusion processes. A key strength is their ability to effectively model data distributions supported on constrained or restricted domains. By adjusting their deterministic dynamics, PDMPs can easily incorporate boundary behaviour, making them straightforward to implement in such settings [Bierkens et al., 2018, Davis, 1993]. Similarly, PDMPs can model data on Riemannian manifolds by employing flows that respect the manifold's geometry (see, e.g., Yang et al. [2022] for a PDMP on the sphere). Moreover, PDMPs are well-suited for modelling data distributions that combine a continuous density and a positive mass on a lower dimensional manifold [Bierkens et al., 2022].

Our contributions are the following:

1) Leveraging the existing literature on time reversals of Markov jump processes [Conforti and Léonard, 2022], we characterise the time reversal of any PDMP under appropriate conditions. It turns out that this time reversal is itself a PDMP with characteristics related to the original PDMP; see Proposition 1.

2) We further specify the characteristics of the time-reversal processes associated with the three aforementioned PDMPs: ZZP, BPS, and RHMC. For these processes, Proposition 2 shows the corresponding time-reversals are PDMPs with simple reversed deterministic motion and with jump rates and kernels that depend on (ratios of) conditional densities of the velocity of the forward process before and after a jump. In contrast to common diffusion models, the emphasis is on distributions of the velocity, similar to the case of the underdamped Langevin diffusion [Dockhorn et al., 2022], which includes an additional velocity vector akin to the PDMPs we consider. Moreover, the structure of the backward jump rates and kernels closely connects to the case of continuous time jump processes on discrete state spaces [Sun et al., 2023, Lou et al., 2024].

3) We define our *piecewise deterministic generative models* employing either ZZP, BPS, or RHMC as forward process, transforming data points to a noise distribution of choice, and develop methodologies to estimate the backward rates and kernels. Then, we define the corresponding backward process based on approximations of the time reversed ZZP, BPS, and RHMC obtained with the estimated rates and kernels. In Section 4 we test our models on simple toy distributions.

4) We obtain a bound for the total variation distance between the data distribution and the distribution of our generative models taking into account two sources of error: first, the approximation of the characteristics of the backward PDMP, and second, its initialisation from the limiting distribution of the forward process; see Theorem 1.

## 2 PDMP based generative models

### 2.1 Piecewise deterministic Markov processes

Informally, a PDMP [Davis, 1984, 1993] on the measurable space $(\mathbb{R}^D, \mathcal{B}(\mathbb{R}^D))$ is a stochastic process that follows deterministic dynamics between random times, while at these times the process can evolve stochastically on the basis of a Markov kernel. In order to define a PDMP precisely, we need three components, called *characteristics* of the PDMP: a *vector field* $\Phi : \mathbb{R}_+ \times \mathbb{R}^D \to \mathbb{R}^D$, which governs the deterministic motion, a *jump rate* $\lambda : \mathbb{R}_+ \times \mathbb{R}^D \to \mathbb{R}_+$, which defines the law of random event times, and finally a *jump kernel* $Q : \mathbb{R}_+ \times \mathbb{R}^D \times \mathcal{B}(\mathbb{R}^D) \to [0,1]$, which is applied at event times and defines the new state of the process. Let us give an informal description of the evolution of a PDMP $Z_t$, clarifying the role of the three characteristics. Suppose at time $T \in \mathbb{R}_+$ the PDMP is at state $z \in \mathbb{R}^D$, that is $Z_T = z$. The deterministic motion of the PDMP is described by the ODE $dZ_{T+s} = \Phi(T + s, Z_{T+s})ds$ for $s \geqslant 0$, with initial condition $Z_T = z$. We introduce the differential flow $\varphi : (t, s, z) \mapsto \varphi_{t,t+s}(z)$, which solves the ODE in the sense that $d\varphi_{t,t+s}(z) = \Phi(T + s, \varphi_{t,t+s}(z))ds$ for $s \geqslant 0$. The process evolves deterministically according to $\varphi$ until the next event time $T + \tau$, where $\tau$ is a random variable with law $\mathbb{P}(\tau > s | Z_T = z) = \exp(-\int_0^s \lambda(\varphi_{T,T+u}(z))du)$, i.e. the exponential distribution with non-homogeneous rate $s \mapsto \lambda(\varphi_{T,T+s}(z))$. We can, at least in principle, simulate $\tau$ by solving

$$\tau = \inf\left\{ t > 0 : \int_0^t \lambda(T + u, \varphi_{T,T+u}(z))du \geqslant E \right\} \tag{1}$$

where $E \sim \mathrm{Exp}(1)$. The process is then defined on $[\mathrm{T}, \mathrm{T} + \tau)$ by $Z_{\mathrm{T}+t} = \varphi_{\mathrm{T},\mathrm{T}+t}(Z_{\mathrm{T}})$ for $t \in [0, \tau)$. At time $\mathrm{T} + \tau$ the process jumps to a new state that is drawn from the Markov kernel $Q$, hence we set $Z_{\mathrm{T}+\tau} \sim Q(\mathrm{T} + \tau, \varphi_{\mathrm{T},\mathrm{T}+\tau}(z), \cdot)$. A realisation of the path of a PDMP for a given time horizon can then be obtained following this procedure (see also Algorithm 1 in Appendix C.1 for a pseudo-code). The formal construction of a PDMP can be found in Appendix A.1.

Typically a PDMP has several types of jumps, which can be represented by a family of jump rates and kernels $(\lambda_i, Q_i)_{i \in \{1,\dots,\ell\}}$. A PDMP of such type can be obtained with the construction we have described by setting

$$\lambda(t, z) = \sum_{i=1}^{\ell} \lambda_i(t, z), \quad Q(t, z, \mathrm{d}z') = \sum_{i=1}^{\ell} \frac{\lambda_i(t, z)}{\lambda(t, z)} Q_i(t, z, \mathrm{d}z'). \tag{2}$$

An alternative, equivalent construction of a PDMP with $\lambda, Q$ satisfying (2) is given in Appendix A.2. Finally, we say a PDMP is homogeneous (as opposed to the non-homogeneous case we have described) when the characteristics do not depend on time, that is $\Phi : \mathbb{R}^D \to \mathbb{R}^D$, $\lambda : \mathbb{R}^D \to \mathbb{R}_+$, and $Q : \mathbb{R}^D \times \mathcal{B}(\mathbb{R}^D) \to [0, 1]$. In all this work, we suppose that the PDMPs that we consider are non-explosive in the sense of Davis [1993], that is they are such that the time of the $n$-th random event goes to $+\infty$ as $n \to +\infty$, almost surely (see Durmus et al. [2021] for conditions ensuring this).

We now introduce the three PDMPs we consider throughout the paper. All these PDMPs are time-homogeneous and live on a state space of the form $\mathsf{E} = \mathbb{R}^d \times \mathsf{V}$, for $\mathsf{V} \subset \mathbb{R}^d$, assuming $V_0 \in \mathsf{V}$. Then, $Z_t$ can be decomposed as $Z_t = (X_t, V_t)$, where $X_t \in \mathbb{R}^d$ is the component of interest and has the interpretation of the position of a particle, whereas $V_t \in \mathsf{V}$ is an auxiliary vector playing the role of the particle's velocity. In the sequel, if there is no risk of confusion, we take the convention that any $z \in \mathbb{R}^d \times \mathsf{V}$, and we write $z = (x, v)$ for $x \in \mathbb{R}^d$ and $v \in \mathsf{V}$. All the PDMPs below have a stationary distribution of the form $\pi(\mathrm{d}x) \otimes \nu(\mathrm{d}v)$, where $\pi$ has density proportional to $x \mapsto \mathrm{e}^{-\psi(x)}$, for $\psi : \mathbb{R}^d \to \mathbb{R}$ a continuously differential potential, and $\nu$ is a simple distribution on $\mathsf{V}$ for the velocity vector. In our experiments we take $\pi$ to be the standard normal distribution, while $\nu$ is the standard normal when $\mathsf{V} = \mathbb{R}^d$ or the uniform distribution when $\mathsf{V}$ is a compact set. Figure 1 shows sample paths for the position vector of the three PDMPs we introduce below.

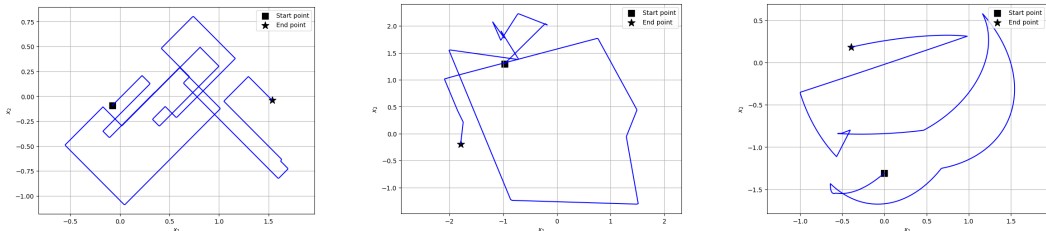

Figure 1: Trace plots for ZZP (left), BPS (centre), RHMC (right). In all cases $\lambda_r = 1$ and $\mathrm{T}_f = 10$.

**The Zig-Zag process** The Zig-Zag process (ZZP) [Bierkens et al., 2019a] is a PDMP with state space $\mathsf{E}^{\mathrm{Z}} = \mathbb{R}^d \times \{-1, 1\}^d$. The deterministic motion is determined by the homogeneous vector field $\Phi^{\mathrm{Z}}(x, v) = (v, 0)^{\mathrm{T}}$, i.e. the particle moves with constant velocity $v$. For $i \in \{1, \dots, d\}$ we define the jump rates $\lambda_i^{\mathrm{Z}}(x, v) := (v_i \partial_i \psi(x))_+ + \lambda_r$, where $(a)_+ = \max(0, a)$, $\partial_i$ denotes the $i$-th partial derivative, and $\lambda_r \geqslant 0$ is a user chosen refreshment rate. The corresponding (deterministic) jump kernels are given by $Q_i^{\mathrm{Z}}((x, v), (\mathrm{d}y, \mathrm{d}w)) = \delta_{(x, \mathscr{R}_i^{\mathrm{Z}} v)}(\mathrm{d}y, \mathrm{d}w)$, where $\delta_z$ denotes the Dirac measure at $z \in \mathsf{E}$. Here, $\mathscr{R}_i^{\mathrm{Z}}$ is the operator that reverses the sign of the $i$-th component of the vector to which it is applied, i.e. $\mathscr{R}_i^{\mathrm{Z}} v = (v_1 \dots, v_{i-1}, -v_i, v_{i+1}, \dots, v_d)$. The ZZP falls within our definition of PDMP taking $\lambda, Q$ as in (2). As shown in Bierkens et al. [2019a], the ZZP has invariant distribution $\pi \otimes \nu$, where $\nu$ is the uniform distribution over $\{\pm 1\}^d$. Moreover, Bierkens et al. [2019b] shows that for any $\lambda_r \geqslant 0$ the law of the ZZP converges exponentially fast to its invariant distribution e.g. when $\pi$ is a standard normal distribution.

**The Bouncy Particle sampler** The Bouncy Particle sampler (BPS) [Bouchard-Côté et al., 2018] is a PDMP with state space is $\mathsf{E}^{\mathrm{B}} = \mathbb{R}^d \times \mathsf{V}^{\mathrm{B}}$, where $\mathsf{V}^{\mathrm{B}} = \mathbb{R}^d$ or $\mathsf{V}^{\mathrm{B}} = \mathsf{S}^{d-1} := \{v \in \mathbb{R}^d : \|v\| = 1\}$.

The deterministic motion is governed as ZZP by the homogeneous vector field defined for $z = (x, v) \in \mathsf{E}$ by $\Phi^{\mathrm{B}}(x, v) = (v, 0)^{\mathrm{T}}$. Now we introduce two jump rates which correspond to two types of random events: *reflections* and *refreshments*. Reflections enforce that $\mu(x, v) = \pi(x)\nu(v)$ is the invariant density of the process, where $\pi(\mathrm{d}x) \propto \exp(-\psi(x))\mathrm{Leb}(\mathrm{d}x)$ is a given distribution and $\nu$ is either a standard normal distribution when $\mathsf{V}^{\mathrm{B}} = \mathbb{R}^d$ or the uniform distribution on $\mathsf{S}^{d-1}$ when $\mathsf{V}^{\mathrm{B}} = \mathsf{S}^{d-1}$. Reflections are associated to the homogeneous jump rate $(x, v) \mapsto \lambda_1^{\mathrm{B}}(x, v) = \langle v, \nabla\psi(x)\rangle_+$, while refreshments are associated to $(x, v) \mapsto \lambda_2^{\mathrm{B}}(x, v) = \lambda_r$ for $\lambda_r > 0$. The corresponding jump kernels are $Q_1^{\mathrm{B}}((x, v), (\mathrm{d}y, \mathrm{d}w)) = \delta_{(x, \mathscr{R}_x^{\mathrm{B}}v)}(\mathrm{d}y, \mathrm{d}w)$ , $Q_2^{\mathrm{B}}((x, v), (\mathrm{d}y, \mathrm{d}w)) = \delta_x(\mathrm{d}y)\nu(\mathrm{d}w)$, where $\mathscr{R}_x^{\mathrm{B}}v = v - 2(\langle v, \nabla\psi(x)\rangle/|\nabla\psi(x)|^2)\nabla\psi(x)$ . The operator $\mathscr{R}_x^{\mathrm{B}}$ *reflects* the velocity $v$ off the hyperplane that is tangent to the contour line of $\psi$ passing though point $x$. The norm of the velocity is unchanged by the application of $\mathscr{R}^{\mathrm{B}}$, and this gives the interpretation that $\mathscr{R}^{\mathrm{B}}$ is an elastic collision of the particle off such hyperplane. As observed in Bouchard-Côté et al. [2018], BPS requires a strictly positive $\lambda_r$ to avoid being reducible, that is to make sure the process can reach any area of the state space. Exponential convergence of the BPS to its invariant distribution was shown by Deligiannidis et al. [2019], Durmus et al. [2020].

**Randomised Hamiltonian Monte Carlo**   Randomised Hamiltonian Monte Carlo (RHMC) [Bou-Rabee and Sanz-Serna, 2017] refers to the PDMP with state space $\mathsf{E}^{\mathrm{H}} = \mathbb{R}^d \times \mathbb{R}^d$ which is characterised by Hamiltonian deterministic flow and refreshments of the velocity vector from the standard normal distribution. The flow is governed by the homogeneous vector field defined by $(x, v) \mapsto \Phi^{\mathrm{H}}(x, v) = (v, -\nabla\psi(x))^{\mathrm{T}}$, where $\psi$ is the potential of $\pi$. The jump rate coincides with the refreshment part of BPS, *i.e.*, it is the constant function $\lambda^{\mathrm{H}} : (x, v) \mapsto \lambda_r > 0$ and jump kernel $Q^{\mathrm{H}}((x, v), (\mathrm{d}y, \mathrm{d}w)) = \delta_x(\mathrm{d}y)\nu(\mathrm{d}w)$. When the stationary distribution $\pi$ is a standard Gaussian, the deterministic dynamics $(x_t, v_t)_{t \geqslant 0}$ satisfy $\mathrm{d}x_t = v_t\mathrm{d}t$, $\mathrm{d}v_t = -x_t\mathrm{d}t$, which for $t \geqslant 0$ has solution $x_t = x_0\cos(t) + v_0\sin(t)$ and $v_t = -x_0\sin(t) + v_0\cos(t)$, where $(x_0, v_0)$ is the initial condition. It is well known that Hamiltonian dynamics preserve the density $\mu(x, v) = \pi(x)\nu(v)$ [Neal, 2010], where $\nu$ is the standard normal distribution, while velocity refreshments are necessary to ensure the process is irreducible. Exponential convergence of the law of this PDMP to $\mu$ was shown in Bou-Rabee and Sanz-Serna [2017].

**Remark 1 (Noise schedule)** *Similarly to diffusion models we can introduce a noise schedule $\beta(t)$ that regulates the amount of randomness injected at time $t$. This can be achieved using the time change of a given PDMP with characteristics $(\Phi, \lambda, Q)$ as forward process, resulting in the PDMP with characteristics $(\Phi_\beta, \lambda_\beta, Q)$ for $\Phi_\beta(t, z) = \beta(t)\Phi(t, z)$ and $\lambda_\beta(t, z) = \beta(t)\lambda(t, z)$.*

## 2.2   Time reversal of PDMPs

In this section we characterise the time reversal of a PDMP. This key result, stated in Proposition 1, is essential to be able to use PDMPs for generative modelling. The *time reversal* of a PDMP $(Z_t)_{t \in [0, \mathrm{T}_f]}$ with initial distribution $\mu_0$ is the process that at time $t \in [0, \mathrm{T}_f]$ has distribution $\mu_0 P_{\mathrm{T}_f-t}$, where $\mu_0 P_s$ denotes the law of $Z_s$. It follows that the law of the time reversal at time $\mathrm{T}_f$ is $\mu_0$, which is the key observation in the context of generative modelling. Characterisations of the law of time reversed Markov processes with jumps were obtained in Conforti and Léonard [2022] and in the following statement we adapt their Theorem 5.7 to our setting, showing that the time reversal of a PDMP with characteristics $(\Phi, \lambda, Q)$ is a PDMP with reversed deterministic motion and jump rates and kernels satisfying (3).

**Proposition 1** *Consider a non-explosive PDMP $(Z_t)_{t \geqslant 0}$ with characteristics $(\Phi, \lambda, Q)$ and initial distribution $\mu_0$ on $\mathbb{R}^D$. In addition, let $\mathrm{T}_f$ be a time horizon. Suppose that $\Phi$ is locally bounded, $(t, z) \mapsto \lambda(t, z)$ is continuous in both its variables, and $\int_0^{\mathrm{T}_f} \mathbb{E}[\lambda(t, Z_t)]\mathrm{d}t < \infty$. Assume the technical conditions **H3**, **H4**, postponed to the appendix. Then, the corresponding time reversal process is a PDMP with characteristics $(\overleftarrow{\Phi}, \overleftarrow{\lambda}, \overleftarrow{Q})$, where $\overleftarrow{\Phi}(t, z) = -\Phi(\mathrm{T}_f - t, z)$ and $\overleftarrow{\lambda}, \overleftarrow{Q}$ are the unique solutions to the following balance equation: for almost all $t \in [0, \mathrm{T}_f]$,*

$$\mu_0 P_{\mathrm{T}_f-t}(\mathrm{d}y)\overleftarrow{\lambda}(t, y)\overleftarrow{Q}(t, y, \mathrm{d}z) = \mu_0 P_{\mathrm{T}_f-t}(\mathrm{d}z)\lambda(\mathrm{T}_f - t, z)Q(\mathrm{T}_f - t, z, \mathrm{d}y) , \qquad (3)$$

*where $\mu_0 P_t$ stands for the distribution of $Z_t$ starting from $\mu_0$.*

The proof is postponed to Appendix A.4. The condition **H3** is standard in the literature on PDMPs [Davis, 1993] and is verified for ZZP, BPS, and RHMC. **H4** is a technical assumption on the do-

main of the generator of the forward PDMP and has been shown to hold e.g. for the ZZP. In the next proposition we derive expressions for the backward jump rate and kernel satisfying (3) corresponding to a forward PDMP with characteristics with the same structure as those of ZZP, BPS, and RHMC. We state the result assuming the PDMP has only one jump type, but the generalisation to the case of $\ell > 1$ jump mechanisms of the form (2) can be immediately obtained applying Proposition 2 to each pair $(\lambda_i, Q_i)$ for $i \in \{1, \ldots, \ell\}$. We refer to Appendix A.6 for the details.

**Proposition 2** *Consider a non-explosive PDMP $(X_t, V_t)_{t \geqslant 0}$ with characteristics $(\Phi, \lambda, Q)$ and initial distribution $\mu_0^X \otimes \mu_0^V$ on $\mathbb{R}^{2d}$. In addition, let $\mathrm{T}_f$ be a time horizon. Suppose that $\Phi$ and $\lambda$ satisfy the same conditions as Proposition 1, in particular the technical conditions **H**3, **H**4 postponed to the appendix. Suppose in addition that for any $t \in (0, \mathrm{T}_f]$, the conditional distribution of $V_t$ given $X_t$ has a transition density $(x, v) \mapsto p_t(v|x)$ with respect to some reference measure $\mu_{\mathrm{ref}}^V$ on $\mathbb{R}^d$.*

*(1) (Deterministic jumps). Suppose $Q((y, w), (\mathrm{d}x, \mathrm{d}v)) = \delta_y(\mathrm{d}x)\delta_{\mathscr{R}_y w}(\mathrm{d}v)$ where for any $y \in \mathbb{R}^d$, $\mathscr{R}_y : \mathbb{R}^d \to \mathbb{R}^d$ is an involution which preserves $\mu_{\mathrm{ref}}^V$, i.e., $\mathscr{R}_y^{-1} = \mathscr{R}_y$ and $\mu_{\mathrm{ref}}^V(\mathrm{d}\mathscr{R}_y w) = \mu_{\mathrm{ref}}^V(\mathrm{d}w)$. Then for almost all $t \in [0, \mathrm{T}_f]$ and any $(y, w) \in \mathbb{R}^{2d}$ such that $p_{\mathrm{T}_f - t}(w|y) > 0$ it holds that*

$$\overleftarrow{\lambda}(t, (y, w)) = \frac{p_{\mathrm{T}_f - t}(\mathscr{R}_y w|y)}{p_{\mathrm{T}_f - t}(w|y)}\lambda(\mathrm{T}_f - t, (y, \mathscr{R}_y w)) , \quad \overleftarrow{Q}((y, w), (\mathrm{d}x, \mathrm{d}v)) = \delta_y(\mathrm{d}x)\delta_{\mathscr{R}_y w}(\mathrm{d}v) .$$

*(2) (Refreshments). Suppose $Q((y, w), (\mathrm{d}x, \mathrm{d}v)) = \delta_y(\mathrm{d}x)\nu(\mathrm{d}v|y)$, where $\nu$ is a transition kernel on $\mathbb{R}^d \times \mathcal{B}(\mathbb{R}^d)$, and $\lambda(t, (y, w)) = \lambda(t, y)$. Suppose also for any $y \in \mathbb{R}^d$, $\nu(\cdot|y)$ is absolutely continuous with respect to $\mu_{\mathrm{ref}}^V$. Then for almost all $t \in [0, \mathrm{T}_f]$ and any $(y, w) \in \mathbb{R}^{2d}$ such that $p_{\mathrm{T}_f - t}(w|y) > 0$ it holds that*

$$\overleftarrow{\lambda}(t, (y, w)) = \frac{(\mathrm{d}\nu/\mathrm{d}\mu_{\mathrm{ref}}^V)(w|y)}{p_{\mathrm{T}_f - t}(w|y)}\lambda(\mathrm{T}_f - t, y), \quad \overleftarrow{Q}(t, (y, w), (\mathrm{d}x, \mathrm{d}v)) = \delta_y(\mathrm{d}x)p_{\mathrm{T}_f - t}(v|x)\mu_{\mathrm{ref}}^V(\mathrm{d}v).$$

The proof is postponed to Appendix A.5. We remark that we consider that $\mu_0^V$ is a distribution on $\mathbb{R}^d$ also when $\mu_0^V(\mathsf{V}) = 1$ for $\mathsf{V} \subset \mathbb{R}^d$, in which case the reference measure can simply be chosen such that $\mu_{\mathrm{ref}}^V(\mathsf{V}) = 1$. Applying Proposition 2 we are able to derive explicit expressions for the characteristics of the time reversals of ZZP, RHMC, and BPS. The rigorous statements and their proofs can be found in Appendix A.7. For ZZP and BPS we assume the following condition on $\pi$, the limiting distribution for the position vector of the forward process.

**H1** *Recall $\pi(x) \propto e^{-\psi(x)}$. It holds that $\psi \in \mathcal{C}^2(\mathbb{R}^d)$ and $\sup_{x \in \mathbb{R}^d} \|\nabla^2 \psi(x)\| < +\infty$.*

This assumption is satisfied e.g. by any multivariate normal distribution. For BPS and RHMC we suppose that for any $t \in (0, \mathrm{T}_f]$, the conditional distribution of $V_t$ given $X_t$ has a transition density $(x, v) \mapsto p_t(v|x)$ with respect to the Lebesgue measure. Moreover, for all samplers we assume **H**4.

**Time reversal of ZZP** In order to apply Proposition 2 we additionally assume that $\int |\partial_i \psi(x)| \mathrm{d}\mu_\star(x) < \infty$ for all $i = 1, \ldots, d$. We find that the deterministic motion is defined by $\overleftarrow{\Phi}^{\mathrm{Z}}(y, w) = (-w, 0)^{\mathrm{T}}$ for any $(y, w) \in \mathbb{R}^{2d}$, while the backward rates and kernels are for $i = 1, \ldots, d$ and for all $(y, w) \in \mathbb{R}^{2d}$ such that $p_{\mathrm{T}_f - t}(w|y) > 0$,

$$\overleftarrow{\lambda}_i^{\mathrm{Z}}(t, (y, w)) = \frac{p_{\mathrm{T}_f - t}(\mathscr{R}_i^{\mathrm{Z}} w|y)}{p_{\mathrm{T}_f - t}(w|y)}\lambda_i^{\mathrm{Z}}(y, \mathscr{R}_i^{\mathrm{Z}} w) , \quad \overleftarrow{Q}_i^{\mathrm{Z}}((y, w), (\mathrm{d}x, v)) = \delta_{(y, \mathscr{R}_i^{\mathrm{Z}} w)}(\mathrm{d}x, v) . \quad (4)$$

**Time reversal of BPS** Whereas in Appendix A.7 we consider the case where the velocity of BPS is initialised on $\mathbb{S}^{d-1}$, we can formally apply Proposition 2 to the case of $\nu$ is the standard $d$-dimensional Gaussian distribution assuming that $\int |\nabla \psi(x)| \mathrm{d}\mu_\star(x) < \infty$. The drift of the backward BPS is clearly the same as for the backward ZZP, while jump rates and kernels are for all $t \in [0, \mathrm{T}_f]$ and $(y, w) \in \mathbb{R}^{2d}$ such that $p_{\mathrm{T}_f - t}(w|y) > 0$

$$\overleftarrow{\lambda}_1^{\mathrm{B}}(t, (y, w)) = \frac{p_{\mathrm{T}_f - t}(\mathscr{R}_y^{\mathrm{B}} w|y)}{p_{\mathrm{T}_f - t}(w|y)}\lambda_1^{\mathrm{B}}(y, \mathscr{R}_y^{\mathrm{B}} w), \quad \overleftarrow{Q}_1^{\mathrm{B}}((y, w), (\mathrm{d}x, \mathrm{d}v)) = \delta_{(y, \mathscr{R}_y^{\mathrm{B}} w)}(\mathrm{d}x, \mathrm{d}v) ,$$

$$\overleftarrow{\lambda}_2^{\mathrm{B}}(t,(y,w)) = \lambda_r \frac{\nu(w)}{p_{\mathrm{T}_f-t}(w|y)} \ , \qquad \overleftarrow{Q}_2^{\mathrm{B}}(t,(y,w),(\mathrm{d}x,\mathrm{d}v)) = p_{\mathrm{T}_f-t}(v|y)\delta_y(\mathrm{d}x)\mathrm{d}v \ . \quad (5)$$

**Time reversal of RHMC.** The deterministic motion of the backward RHMC follows the system of ODEs $\overleftarrow{\Phi}^{\mathrm{H}}(x,v) = (-v,\nabla\psi(x))^{\mathrm{T}}$, which, when the limiting distribution $\pi$ is Gaussian, has solution $x_t = x_0\cos(t) - v_0\sin(t)$ and $v_t = x_0\sin(t) + v_0\cos(t)$. The backward refreshment rate and kernel coincide with those of BPS as given in (5).

**Remark 2 (Variance exploding PDMPs)** *Similarly to the case of diffusion models [Song et al., 2021], we can define* variance exploding PDMPs *choosing $\psi(x) = 0$ for all $x \in \mathbb{R}^d$, that is when $\pi(\mathrm{d}x)$ is the Lebesgue measure. In this case, the deterministic motion of RHMC coincides with ZZP and BPS, and all three processes have only velocity refreshment events.*

### 2.3 Approximating the characteristics of time reversals of PDMPs

In Section 2.2 we showed that the backward jump rates and kernels of ZZP, BPS, and RHMC, involve the conditional densities of the velocity vector of the forward process given its position vector at all times $t \in [0,\mathrm{T}_f]$. These conditional densities are unavailable in analytic form, hence in this section we develop methods to learn the jump rates and kernels of our time reversed PDMPs. In Appendix D we give the pseudo codes and more detailed descriptions of the training procedure for our models, together with a comparison with diffusion models.

**Approximating the jump rates of the backward ZZP via ratio matching** In the case of ZZP, we need to approximate for any $i \in \{1,\ldots,d\}$, the rates in (4). Since the terms $\lambda_i^{\mathrm{Z}}(x,\mathscr{R}_i^{\mathrm{Z}}v)$ are known, it is sufficient to estimate the density ratios $r_i^{\mathrm{Z}}(x,v,t) := {p_t(\mathscr{R}_i^{\mathrm{Z}}v|x)}/{p_t(v|x)}$ for all states $(x,v)$ such that $p_t(v|x) > 0$. To this end, we introduce a class of functions $\{s^\theta : \mathbb{R}^d \times \{-1,1\}^d \times [0,\mathrm{T}_f] \to \mathbb{R}_+^d \ : \ \theta \in \Theta\}$ for some parameter set $\Theta \subset \mathbb{R}^{d_\theta}$ and aim to find a parameter $\theta_\star \in \Theta$ such that for any $i \in \{1,\ldots,d\}$, the $i$-th component of $s^{\theta_\star}$, denoted by $s_i^{\theta_\star}(\cdot)$, is an approximation of $r_i^{\mathrm{Z}}$. We then approximate the backward ZZP by using the rates $\bar{\lambda}_i^{\mathrm{Z}}(t,(x,v)) = s_i^{\theta_\star}(x,v,\mathrm{T}_f-t)\,\lambda_i^{\mathrm{Z}}(x,\mathscr{R}_i^{\mathrm{Z}}v)$. To address the problem of fitting $\theta$, we consider different loss functions inspired by the ratio matching (RM) problem considered in Hyvärinen [2007].

From a discrete probability density $p_\pm$ on $\{-1,1\}^d$, RM consists in learning the $d$ ratios $v \mapsto {p_\pm(\mathscr{R}_i v)}/{p_\pm(v)}$ for $i \in \{1,\ldots,d\}$. This problem was motivated in Hyvärinen [2007] as a means to estimate $p_\pm$ without requiring its normalising constant, similarly to score matching applied to estimate continuous probability densities [Hyvärinen, 2005]. In our context we are interested only in the ratios, hence as opposed to Hyvärinen [2007] we do not model the conditional distributions $(x,v) \mapsto p_t(v|x)$, but directly the ratios $r_i^{\mathrm{Z}}$. Adapting the ideas of Hyvärinen [2007] to our context, we introduce the function $\mathbf{G} : r \mapsto (1+r)^{-1}$ and define the *Explicit Ratio Matching* objective function

$$\ell_{\mathrm{E}}(\theta) = \int_0^{\mathrm{T}_f} \mathrm{d}t\,\omega(t) \sum_{i=1}^d \mathbb{E}\Big[\{\mathbf{G}(s_i^\theta(X_t,V_t,t)) - \mathbf{G}(r_i(X_t,V_t,t))\}^2$$
$$+ \{\mathbf{G}(s_i^\theta(X_t,\mathscr{R}_i^{\mathrm{Z}}V_t,t)) - \mathbf{G}(r_i(X_t,\mathscr{R}_i^{\mathrm{Z}}V_t,t))\}^2\Big] \ . \quad (6)$$

where $\omega : [0,\mathrm{T}_f] \to \mathbb{R}_+^*$ is a probability density, and $(X_t,V_t)_{t\geqslant 0}$ is a ZZP initialised from $\mu_\star \otimes \mathrm{Unif}(\{-1,1\}^d)$. This objective function considers simultaneously the square error in the estimation of both $(x,v,t) \mapsto r_i(x,v,t)$ and $(x,v,t) \mapsto r_i(x,\mathscr{R}_i^{\mathrm{Z}}v,t)$, where the function $\mathbf{G}$ improves numerical stability, particularly when one of the two ratios is very small. Clearly $\ell_{\mathrm{E}}(\theta) = 0$ if and only if $s_i^\theta(x,v,t) = r_i(x,v,t)$ for almost all $x,v,t$ and all $i$. Moreover, the choice of $\mathbf{G}$ allows us to optimise without knowledge of the true ratios, as shown in the following result.

**Proposition 3** *It holds that* $\arg\min_\theta \ell_{\mathrm{E}}(\theta) = \arg\min_\theta \ell_{\mathrm{I}}(\theta)$ *for*

$$\ell_{\mathrm{I}}(\theta) = \int_0^{\mathrm{T}_f} \mathrm{d}t\,\omega(t) \sum_{i=1}^d \mathbb{E}\Big[\mathbf{G}^2(s_i^\theta(X_t,V_t,t)) + \mathbf{G}^2(s_i^\theta(X_t,\mathscr{R}_i^{\mathrm{Z}}V_t,t)) - 2\mathbf{G}(s_i^\theta(X_t,V_t,t))\Big] \ , \quad (7)$$

*where $(X_t,V_t)_{t\in\mathbb{R}_+}$ is a ZZP starting from $\mu_\star \otimes \mathrm{Unif}(\{-1,1\}^d)$.*

Therefore we aim to solve the minimisation problem associated with $\ell_{\mathrm{I}}$, which has for empirical counterpart

$$\theta \mapsto \frac{1}{N} \sum_{n=1}^{N} \sum_{i=1}^{d} \mathbf{G}^2(s_i^\theta(X_{\tau^n}^n, V_{\tau^n}^n, \tau^n)) + \mathbf{G}^2(s_i^\theta(X_{\tau^n}^n, \mathscr{R}_i^Z V_{\tau^n}^n, \tau^n)) - 2\mathbf{G}(s_i^\theta(X_{\tau^n}^n, V_{\tau^n}^n, \tau^n))$$

where $\{\tau^n\}_{n=1}^N$ are i.i.d. samples from $\omega$, independent of $\{(X_t^n, V_t^n)_{t \geqslant 0}\}_{n=1}^N$, which are $N$ i.i.d. realisations of the ZZP respectively starting at the $n$-th training data point with velocity $V_0^n$, where $\{V_0^n\}_{n=1}^N$ are i.i.d. observations of $\mathrm{Unif}(\{-1,1\}^d)$.

Notice that the loss above has computational cost increasing linearly in $d$ because $d+1$ evaluations of the model are needed for each datum. This can be improved considering an estimate for the ratio which does not take as input the whole velocity vector (see Appendix D.1 for the details). This variation has computational cost that is constant in the dimension, but might have lower accuracy.

**Approximating the characteristics of BPS and RHMC** For BPS and RHMC, Proposition 2 shows that if we aim to sample from the backward process, we have to estimate both ratios of the conditional density of the velocity of the forward PDMP given its position at any time $t \in [0, \mathrm{T}_f]$, and also to be able to sample from such densities as prescribed by the backward jump kernel (5). In order to address both requirements, we introduce a parametric family of conditional probability distributions $\{p_\theta : \theta \in \Theta\}$ of the form $(x, v, t) \mapsto p_\theta(v|x, t)$, where $\Theta \subset \mathbb{R}^{d_\theta}$, which we model with the framework of normalising flows (NFs) [Papamakarios et al., 2021]. The advantage of NFs lays in their feature that, once the network is learned, it is possible both to obtain an estimate of the density at a given state and time, and also to generate samples which are approximately from $(x, v, t) \mapsto p_t(v|x)$. However, training conditional NFs can be challenging in high dimensions.

Focusing on BPS, we now illustrate how we can use NFs to learn the backward jump rates and kernels. We aim to find a parameter $\theta_\star^{\mathrm{B}}$ such that $p_{\theta_\star^{\mathrm{B}}}(v|x, t)$ is a good approximation of $p_t(v|x)$, the conditional density of the forward BPS with respect to the Lebesgue measure. We choose to optimise $\theta$ following the maximum likelihood approach, which gives the theoretical loss

$$\ell_{\mathrm{ML}}(\theta) = -\int_0^{\mathrm{T}_f} dt\, \omega(t) \mathbb{E}\left[\log p_\theta(V_t|X_t, t)\right], \tag{8}$$

where $\omega : [0, \mathrm{T}_f] \to \mathbb{R}_+^*$ is a probability density, and $(X_t, V_t)_{t \geqslant 0}$ is a a BPS initialised from $\mu_\star \otimes \nu$, with $\nu$ denoting the density of the $d$-dimensional standard normal distribution. The optimal parameter $\theta_\star^{\mathrm{B}}$ can then be found minimising the empirical counterpart of $\ell_{\mathrm{ML}}(\theta)$:

$$\theta_\star^{\mathrm{B}} = \arg\min_\theta \frac{1}{N} \sum_{n=1}^{N} \log p_\theta(V_{\tau^n}^n | X_{\tau^n}^n, \tau^n), \tag{9}$$

where $\{\tau^n\}_{n=1}^N$ are i.i.d. samples from $\omega$, independent of $\{(X_t^n, V_t^n)_{t \geqslant 0}\}_{n=1}^N$, which are $N$ i.i.d. realisations of the ZZP respectively starting at the $n$-th training data point with velocity $V_0^n$, where $\{V_0^n\}_{n=1}^N$ are i.i.d. observations from the multivariate standard normal distribution. Once we have obtained the optimal parameter $\theta_\star^{\mathrm{B}}$, we can define our approximation of the backward refreshment mechanism of BPS taking the rate $\bar{\lambda}_2^{\mathrm{B}}(t, (x, v)) = \lambda_r \times \nu(v)/p_{\theta_\star^{\mathrm{B}}}(v|x, \mathrm{T}_f - t)$ and the kernel $\bar{Q}_2^{\mathrm{B}}(t, (y, w), (dx, dv)) = p_{\theta_\star^{\mathrm{B}}}(v|y, \mathrm{T}_f - t)\delta_y(dx)dv$. Similarly, we estimate the backward reflection ratio of BPS as $\bar{\lambda}_1^{\mathrm{B}}(t, (x, v)) = \lambda_1^{\mathrm{B}}(x, \mathscr{R}_x^{\mathrm{B}} v) \times p_{\theta_\star^{\mathrm{B}}}(\mathscr{R}_x^{\mathrm{B}} v|x, \mathrm{T}_f - t)/p_{\theta_\star^{\mathrm{B}}}(v|x, \mathrm{T}_f - t)$.

## 2.4 Simulating the backward process

We now discuss how we can simulate the backward PDMP with exact backward flow map $(t, x, v) \mapsto \varphi_{-t}(x, v)$ and jump characteristics $\bar{\lambda}$ and $\bar{Q}$ that are approximations of the jump rates and kernels of the time reversed PDMPs obtained as discussed in Section 2.3. We recall that the backward rates have the general form $\bar{\lambda}(t, (x, v)) = s_\theta(x, v, \mathrm{T}_f - t)\lambda(x, \mathscr{R}v)$, where $s_\theta$ is an estimate of a density ratio and $\mathscr{R}$ is a suitable involution. In principle such a PDMP can be simulated following the procedure described in Section 2.1, but the generation of the random jump times via (1) requires the integration of $\bar{\lambda}(t, \varphi_{-t}(x, v))$ with respect to $t$. This cannot be achieved since $\bar{\lambda}$ is defined through a neural network. A standard approach in the literature (see e.g. Bertazzi et al.

[2022, 2023]) is to discretise time and (informally) approximate the integral in (1) with a finite sum. Here we focus on approximations based on splitting schemes discussed in Bertazzi et al. [2023], adapting their ideas to the non-homogeneous case. Such splitting schemes approximate a PDMP with a Markov chain defined on the time grid $\{t_n\}_{n\in\{0,...,N\}}$, with $t_0 = 0$ and $t_N = \mathrm{T}_f$. The key idea is that the deterministic motion and the jump part of the PDMP are simulated separately in a suitable order, obtaining second order accuracy under suitable conditions (see Theorem 2.6 in Bertazzi et al. [2023]). Now, we give an informal description of the splitting scheme that we use for RHMC, that is based on splitting DJD in Bertazzi et al. [2023], where D stands for deterministic motion and J for jumps. We define our Markov chain based on the step sizes $\{\delta_j\}_{j\in\{1,...,N\}}$, where $\delta_j = t_j - t_{j-1}$. Suppose we have defined the Markov chain on $\{t_k\}_{k\in\{0,...,n\}}$ for $n < N$ and that the state at time $t_n$ is $(x_{t_n}, v_{t_n})$. The next state is obtained following three steps. *First*, the particle moves according to its deterministic motion for a half-step, that is we define an intermediate state $(\tilde{x}_{t_n}, \tilde{v}_{t_n}) = \varphi_{-\delta_{n+1}/2}(x_{t_n}, v_{t_n})$. *Second*, we turn our attention to the jump part of the process. In this phase, the particle is only allowed to move through jumps and there is no deterministic motion. This means that the rate is frozen to the value $\overline{\lambda}(t_n + {}^{\delta_{n+1}}\!/2, (\tilde{x}_{t_n}, \tilde{v}_{t_n}))$ and thus the integral in (1) can be computed trivially. The proposal for the next event time is then given by $\tau_{n+1} \sim \mathrm{Exp}(\overline{\lambda}(t_n + \delta_{n+1}/2, (\tilde{x}_{t_n}, \tilde{v}_{t_n})))$. If $\tau_{n+1} \leqslant \delta_{n+1}$, we draw $w \sim \overline{Q}(t_n + \delta_{n+1}/2, (\tilde{x}_{t_n}, \tilde{v}_{t_n}), \cdot)$ and update $\tilde{v}_{t_n} = w$, else we leave $\tilde{v}_{t_n}$ unchanged. *Finally* we conclude with an additional half-step of deterministic motion, letting $(x_{t_{n+1}}, v_{t_{n+1}}) = \varphi_{-\delta_{n+1}/2}(\tilde{x}_{t_n}, \tilde{v}_{t_n})$. We refer to Appendix C.2 for a detailed description of the schemes used for each process together with the pseudo-codes.

## 3 Error bound in total variation distance

In this section, we give a bound on the total variation distance between the data distribution $\mu_\star$ and the law of the synthetic data generated by a PDMP with initial distribution $\pi \otimes \nu$ and approximate characteristics obtained, e.g., with the methods described in Section 2.3. We obtain our result comparing the law of such PDMP to the law of the exact time reversal obtained in Section 2.2, that is the PDMP with the analytic characteristics of Proposition 2 and with initial distribution $\mathcal{L}(X_{\mathrm{T}_f}, V_{\mathrm{T}_f})$, i.e. the law of the forward PDMP at time $\mathrm{T}_f$ when initialised from $\mu_\star \otimes \nu$. In Theorem 1 below we then take into account two of the three sources of error in our models: (i) the error introduced initialising the backward PDMP from the limiting distribution of the forward, (ii) the error due to the approximation of the backward rates and kernels. For simplicity we neglect the discretisation error caused by the methods discussed in Section 2.4.

We shall assume the following condition, which deals with the error introduced by initialising the backward PDMP from $\pi \otimes \nu$.

**H2** *The forward PDMP with semigroup $(P_t)_{t\geqslant 0}$ is such that there exist $\gamma, C > 0$ for which*
$$\|\pi \otimes \nu - \mu_\star \otimes \nu P_t\|_{\mathrm{TV}} \leqslant Ce^{-\gamma t}.$$

Informally, **H**2 is verified for some $C < \infty$ when $\pi$ is a multivariate standard Gaussian distribution for ZZP and BPS if the tails of $\mu_\star$ are at least as light as those of $\pi$, while for RHMC it is enough if $\mu_\star$ has finite second moments. We refer to Appendix E.1 for a more detailed discussion on this aspect. We are now ready to state our result.

**Theorem 1** *Consider a non-explosive PDMP $(X_t, V_t)_{t\geqslant 0}$ with initial distribution $\mu_\star \otimes \nu$, stationary distribution $\pi \otimes \nu$, and characteristics $(\Phi, \lambda, Q)$. Let $\mathrm{T}_f$ be a time horizon. Suppose the assumptions of Proposition 1 as well as **H**2 hold. Let $(\overline{X}_t, \overline{V}_t)_{t\in[0,\mathrm{T}_f]}$ be a non-explosive PDMP initial distribution $\pi \otimes \nu$ and characteristics $(\overline{\Phi}, \overline{\lambda}, \overline{Q})$, where $\overline{\Phi}(t, (x, v)) = \Phi(\mathrm{T}_f - t, (x, v))$ for all $t \in [0, \mathrm{T}_f]$ and $(x, v) \in \mathbb{R}^{2d}$. Then it holds that*

$$\|\mu_\star - \mathcal{L}(\overline{X}_{\mathrm{T}_f})\|_{\mathrm{TV}} \leqslant Ce^{-\gamma \mathrm{T}_f} + 2\mathbb{E}\left[1 - \exp\left(-\int_0^{\mathrm{T}_f} g_{\mathrm{T}_f - t}(X_t, V_t)\mathrm{d}t\right)\right], \qquad (10)$$

*where*

$$g_t(x, v) = \frac{(\overleftarrow{\lambda} \wedge \overline{\lambda})(t, (x, v))}{2}\|\overleftarrow{Q}(t, (x, v), \cdot) - \overline{Q}(t, (x, v), \cdot)\|_{\mathrm{TV}} + \left|\overleftarrow{\lambda}(t, (x, v)) - \overline{\lambda}(t, (x, v))\right| \tag{11}$$

*and $\overleftarrow{\lambda}, \overleftarrow{Q}$ are as given by Proposition 1.*

| Dataset | i-DDPM | BPS | RHMC | ZZP |
|---|---|---|---|---|
| Checkerboard | $2.49 \pm 0.98$ | $1.96 \pm 1.51$ | $4.27 \pm 3.36$ | $\mathbf{0.81} \pm 0.19$ |
| Fractal tree | $8.04 \pm 5.58$ | $2.25 \pm 1.70$ | $4.41 \pm 4.35$ | $\mathbf{1.12} \pm 0.58$ |
| Gaussian grid | $23.19 \pm 9.72$ | $4.59 \pm 4.03$ | $\mathbf{4.01} \pm 3.32$ | $4.43 \pm 4.05$ |
| Olympic rings | $2.03 \pm 1.60$ | $2.07 \pm 1.19$ | $2.41 \pm 2.24$ | $\mathbf{1.43} \pm 0.86$ |
| Rose | $6.77 \pm 5.81$ | $1.92 \pm 1.57$ | $2.16 \pm 1.59$ | $\mathbf{0.90} \pm 0.35$ |

Table 1: MMD $\downarrow$, in units of $1e-3$, averaged over 6 runs, with the corresponding standard deviations.

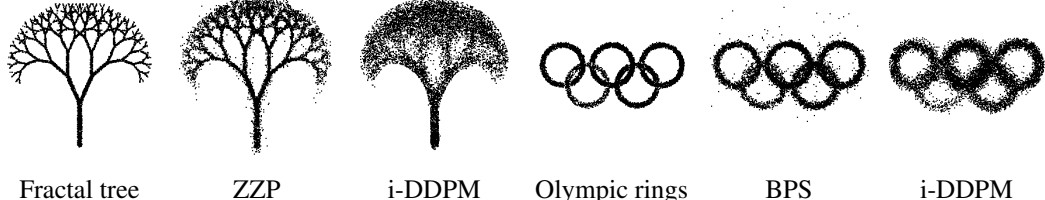

Fractal tree     ZZP     i-DDPM     Olympic rings     BPS     i-DDPM

Figure 2: Comparative results on two-dimensional generation of synthetic datasets.

The proof is postponed to Appendix E.2. The first term in (10) is caused by initialising the process from $\pi \otimes \nu$, while the second term represents the error introduced by the approximate jump rate $\overline{\lambda}$ and kernel $\overline{Q}$. For the sake of illustration we obtain a simple upper bound to (10) in the case of ZZP (for the details see Appendix E.3). We assume the conditions of Theorem 1 are satisfied and also that the expected error of the learned rates $\bar{\lambda}_i^Z$ is bounded by a constant, in the sense that $\mathbb{E}[|r_i^Z(X_t, V_t, \mathrm{T}_f - t) - s_i^\theta(X_t, V_t, \mathrm{T}_f - t)|\lambda_i^Z(X_t, \mathscr{R}_i^Z V_t)] \leqslant M$ for all $t \in [0, \mathrm{T}_f]$ and $i \in \{1, \ldots, d\}$. The latter condition is similar to the standard assumption asked on the approximation of the score in diffusion models [Chen et al., 2023]. Under these conditions we obtain the bound

$$\|\mu_\star - \mathcal{L}(\overline{X}_{\mathrm{T}_f})\|_{\mathrm{TV}} \leqslant C e^{-\gamma \mathrm{T}_f} + 4M\mathrm{T}_f d . \tag{12}$$

## 4 Numerical simulations

In this section, we test our piecewise deterministic generative models on simple synthetic datasets.

**Design** We compare the generative models based on ZZP, BPS, and RHMC with the improved denoising diffusion probabilistic model (i-DDPM) given in Nichol and Dhariwal [2021]. For all of our models, we choose the standard normal distribution as target distribution for the position vector, as well as for the velocity vector in the cases of BPS and RHMC. The accuracy of trained generative models is evaluated by the kernel maximum mean discrepancy (MMD). We refer to Appendix F for a detailed description of the parameters and networks choices.

**Sample quality** In Table 1 we report the MMD score for five, 2-dimensional toy distributions. We observe that the PDMP based generative models perform well compared to i-DDPM in all of these five datasets. In particular, ZZP and i-DDPM are implemented with the same neural network architecture, hence ZZP appears to compare favourably to i-DDPM with the same model expressivity. The results of Table 1 are supported by the plots of generated data shown in Figure 2, illustrating how ZZP and BPS are able to generate more detailed edges compared to i-DDPM.

In Figure 4, we compare the output of RHMC and i-DDPM for a very small number of reverse steps. We observe how in this setting the data generated by RHMC are noticeably closer to the true data distribution compared to i-DDPM. This phenomenon is observed also for BPS as shown in Table 2, and is intuitively caused by the refreshment kernel, which is able to generate velocities that correct wrong positions. Respecting this intuition, ZZP does not perform as well as BPS and RHMC for a small number of reverse steps since its velocities are constrained to $\{-1, 1\}$. Nonetheless, ZZP generates the most accurate results in our experiments given a large enough number of reverse steps.

Table 2 and Figure 3 show that PDMP-based models require a smaller computational time to generate samples of a given quality compared to i-DDPM. This is the case because PDMP models require

considerably less backward steps than i-DDPM, although each step is more expensive (see Table 2). Additional results can be found in Appendix F, including promising results applying ZZP to the MNIST dataset.

Table 2: MMD ↓ for various number of backward steps, Rose dataset.

| steps | i-DDPM | BPS | RHMC | ZZP |
|---|---|---|---|---|
| 2 | 696.28 | 165.09 | **26.48** | 358.25 |
| 5 | 192.17 | 22.18 | **3.00** | 89.49 |
| 10 | 45.08 | 5.48 | **1.75** | 11.31 |
| 25 | 12.34 | 1.58 | **0.60** | 1.20 |
| 100 | 8.72 | 3.66 | 1.72 | **1.04** |
| time/step(ms) | 3.94 | 45.8 | 15.1 | 11.2 |

Figure 3: MMD ↓, runtime (ms) per method, Rose dataset.

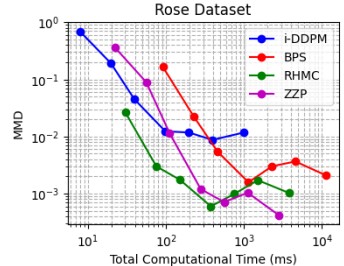

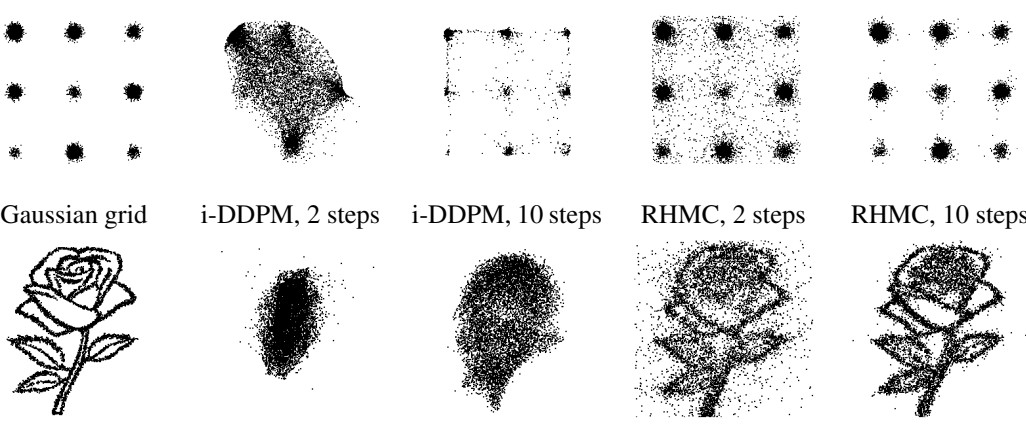

| Gaussian grid | i-DDPM, 2 steps | i-DDPM, 10 steps | RHMC, 2 steps | RHMC, 10 steps |
| Rose dataset | i-DDPM, 2 steps | i-DDPM, 10 steps | RHMC, 2 steps | RHMC, 10 steps |

Figure 4: Comparing RHMC and i-DDPM for small number of reverse steps.

## 5 Discussion and conclusions

We have introduced new generative models based on piecewise deterministic Markov processes, developing a theoretically sound framework with specific focus on three PDMPs from the sampling literature. While this work lays the foundations of this class of methods, it also opens several directions worth investigating in the future.

Similarly to other generative models, our PDMP based algorithms are sensitive to the choice of the network architecture that is used to approximate the backward characteristics. Therefore, it is crucial to investigate which architectures are most suited for our algorithms in order to achieve state of the art performance in real world scenarios. For instance, in the case of BPS and RHMC it could be beneficial to separate the estimation of the density ratios and the generation of draws of the velocity conditioned on the position and time. For the case of ZZP, efficient techniques to learn the network in a high dimensional setting need to be investigated, while network architectures that resemble those used to approximate the score function appear to adapt well to the case of density ratios. Moreover, there are several alternative PDMPs that could be used as generative models and that we did not consider in detail in this paper, as for instance variance exploding alternatives.

## Acknowledgments and Disclosure of Funding

AB, EM, AD are funded by the European Union (ERC-2022-SyG, 101071601). US and DS are funded by the European Union (ERC, Dynasty, 101039676). Views and opinions expressed are however those of the authors only and do not necessarily reflect those of the European Union or the European Research Council Executive Agency. Neither the European Union nor the granting authority can be held responsible for them. US is additionally funded by the French government under management of Agence Nationale de la Recherche as part of the "Investissements d'avenir" program, reference ANR-19-P3IA-0001 (PRAIRIE 3IA Institute). The authors are grateful to the CLEPS infrastructure from the Inria of Paris for providing resources and support.

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

# Appendix

The Appendix is organised as follows. Appendix A includes the details and the proofs regarding Section 2.1 and Section 2.2. Appendix B contains details and proofs regarding the framework of density ratio matching. Appendix C gives details and pseudo-codes for the exact simulation of our forward processes (see Appendix C.1), and also for the splitting schemes that are used to approximate the backward processes (see Appendix C.2). Appendix D details the algorithms used to train our generative models and outlines the computational differences with the popular framework of denoising diffusion models. Appendix E contains the proof for Theorem 1 and some related details Finally, Appendix F contains the details on the numerical simulations, as well as additional results.

## A   PDMPs and their time reversals

### A.1   Construction of a PDMP

Here we describe the formal construction of a PDMP with the characteristics $(\Phi, \lambda, Q)$. To this end, consider the differential flow $\varphi : (t, s, z) \mapsto \varphi_{t,t+s}(z)$, which solves the ODE, $\mathrm{d}z_{t+s} = \Phi(t+s, z_{t+s})\mathrm{d}s$ for $s \geqslant 0$, i.e. $z_{t+s} = \varphi_{t,t+s}(z_t)$. We define by recursion on $n \in \mathbb{N}$ the process on $(Z_t)_{t \in [0, \mathrm{T}_n]}$ on $[0, \mathrm{T}_n]$ and the increasing sequence of jump times $(T_n)_{n \in \mathbb{N}}$ starting from an initial state $Z_0$ and setting $\mathrm{T}_0 = 0$. Assume that $(\mathrm{T}_i)_{i \in \{0,\ldots,n\}}$ and $(Z_t)_{t \in [0, \mathrm{T}_n]}$ are defined for some $n \in \mathbb{N}$. We now define $(Z_t)_{t \in [\mathrm{T}_n, \mathrm{T}_{n+1}]}$. First, we define

$$\tau_{n+1} = \inf \left\{ t > 0 : \int_0^t \lambda(T_n + u, \varphi_{\mathrm{T}_n, \mathrm{T}_n + u}(Z_{\mathrm{T}_n}))\mathrm{d}u \geqslant E_{n+1} \right\} \tag{13}$$

where $E_{n+1} \sim \mathrm{Exp}(1)$, and set the $n+1$-th jump time $\mathrm{T}_{n+1} = \mathrm{T}_n + \tau_{n+1}$. The process is then defined on $[\mathrm{T}_n, \mathrm{T}_{n+1})$ by $Z_{\mathrm{T}_n + t} = \varphi_{\mathrm{T}_n, \mathrm{T}_n + t}(Z_{\mathrm{T}_n})$ for $t \in [0, \tau_{n+1})$. Finally, we set $Z_{\mathrm{T}_{n+1}} \sim Q(\mathrm{T}_{n+1}, \varphi_{\mathrm{T}_n, \mathrm{T}_n + \tau_{n+1}}(Z_{\mathrm{T}_n}), \cdot)$. The process $(Z_t)_{t \geqslant 0}$ is a Markov process by [Jacobsen, 2005, Theorem 7.3.1].

### A.2   Construction of a PDMP with multiple jump types

In this section we describe the formal construction of a non-homogeneous PDMP with the characteristics $(\Phi, \lambda, Q)$ where $\lambda, Q$ are of the form

$$\lambda(t, z) = \sum_{i=1}^{\ell} \lambda_i(t, z) , \quad Q(t, z, \mathrm{d}z') = \sum_{i=1}^{\ell} \frac{\lambda_i(t, z)}{\lambda(t, z)} Q_i(t, z, \mathrm{d}z') . \tag{14}$$

Recall the differential flow $\varphi : (t, s, z) \mapsto \varphi_{t,t+s}(z)$, which solves the ODE, $\mathrm{d}z_{t+s} = \Phi(t+s, z_{t+s})\mathrm{d}s$ for $s \geqslant 0$, i.e. $z_{t+s} = \varphi_{t,t+s}(z_t)$. Similarly to the case of one type of jump only, we start the PDMP from an initial state $Z_0$, assume it is defined as $(Z_t)_{t \in [0, \mathrm{T}_n]}$ on $[0, \mathrm{T}_n]$ for some $n \in \mathbb{N}$, and we now define define $(Z_t)_{t \in [\mathrm{T}_n, \mathrm{T}_{n+1}]}$. First, we define the proposals $(\tau_{n+1}^i)_{i \in \{1,\ldots,\ell\}}$ for next event time as

$$\tau_{n+1}^i = \inf \left\{ t > 0 : \int_0^t \lambda_i(T_n + u, \varphi_{\mathrm{T}_n, \mathrm{T}_n + u}(Z_{\mathrm{T}_n}))\mathrm{d}u \geqslant E_{n+1}^i \right\}$$

where $E_{n+1}^i \sim \mathrm{Exp}(1)$ for $i \in \{1, \ldots, \ell\}$. Then define $i^* = \arg\min_{i \in \{1,\ldots,\ell\}} \tau_{n+1}^i$ and set the next jump time to

$$T_{n+1} = T_n + \tau_{n+1}^{i^*}.$$

The process is then defined on $[\mathrm{T}_n, \mathrm{T}_{n+1})$ by $Z_{\mathrm{T}_n + t} = \varphi_{\mathrm{T}_n, \mathrm{T}_n + t}(Z_{\mathrm{T}_n})$ for $t \in [0, \tau_{n+1})$. Finally, we set $Z_{\mathrm{T}_{n+1}} \sim Q_{i^*}(T_{n+1}, \varphi_{\mathrm{T}_n, \mathrm{T}_n + \tau_{n+1}}(Z_{\mathrm{T}_n}), \cdot)$.

### A.3   Extended generator

In order to obtain the generator of a PDMP, Theorem 26.14 of Davis [1993] requires "standard conditions" on the characteristics (see conditions (24.8) in Davis [1993]). We state these conditions for a non-homogeneous PDMP in the next assumption.

**H3** *The non-homogeneous characteristics* $(\Phi, \lambda, Q)$ *satisfy the following conditions:*

1. *$\Phi$ is locally Lipschitz and the associated flow map $\varphi$ has infinite explosion time;*

2. *$\lambda$ is such that $u \mapsto \lambda(\varphi_{t,t+u}(x))$ is integrable on $[0, \varepsilon(x,t))$ for some $\varepsilon(x,t) > 0$ and all $(t,x) \in \mathbb{R}_+ \times \mathrm{E}$.*

3. *$Q$ is measurable and such that $Q(t, x, \{x\}) = 0$ for all $(t,x) \in \mathbb{R}_+ \times \mathrm{E}$.*

4. *Let $(\mathrm{T}_n)_{n \in \{0,1,\dots\}}$ be the random sequence of event times of the PDMP and define $N_t = \sum_{k=0}^{\infty} \mathbb{1}_{t \geqslant \mathrm{T}_k}$. It holds that $\mathbb{E}_x[N_t] < \infty$ for all $(t,x) \in \mathbb{R}_+ \times \mathrm{E}$.*

Notably, the PDMP is required to be non-explosive in the sense that the expected number of random events after any time $t$ starting the PDMP from any state should be finite. These conditions are verified for all the three PDMPs we consider as forward processes. Assuming **H3** we can apply Theorem 26.14 in Davis [1993] to the homogeneous PDMP obtained including the time variable, which gives that the extended generator of the non-homogeneous PDMP with characteristics $(\Phi, \lambda, Q)$ is given by

$$\mathscr{L}_t f(z) = \langle \Phi(t,z), \nabla_z f(z) \rangle + \lambda(t,z) \int_{\mathbb{R}^d} (f(y) - f(z)) Q(t, z, \mathrm{d}y), \tag{15}$$

for all functions $f \in \mathrm{dom}(\mathscr{L}_t)$, that is the space of measurable functions such that

$$M_t^f = f(Z_t) - f(Z_0) - \int_0^t \mathscr{L}_s f(Z_s) \mathrm{d}s$$

is a local martingale. We also introduce the Carré du champ $\Gamma_t(f,g) := \mathscr{L}_t(fg) - f\mathscr{L}_t g - g\mathscr{L}_t f$, with domain $\mathrm{dom}(\Gamma_t) := \{f, g : f, g, fg \in \mathrm{dom}(\mathscr{L}_t)\}$ which in the case of a PDMP with generator (15) takes the form

$$\Gamma_t(f,g)(z) = \lambda(t,z) \int_{\mathbb{R}^d} (f(y) - f(z))(g(y) - g(z)) Q(t, z, \mathrm{d}y).$$

### A.4  Proof of Proposition 1

In order to prove Proposition 1 we apply Conforti and Léonard [2022, Theorem 5.7] and hence in this section we verify the required assumptions. Before starting, we state the following technical condition which we omitted in Proposition 1 and is assumed in Conforti and Léonard [2022, Theorem 5.7].

**H4** *It holds $\mathrm{C}_c^2(\mathbb{R}^d) \subset \mathrm{dom}(\mathscr{L}_t)$ for any $t \in \mathbb{R}_+$.*

We now turn to verifying the remaining assumptions in Conforti and Léonard [2022, Theorem 5.7]. The "General Hypotheses" of Conforti and Léonard [2022] are satisfied since we assume the vector field $\Phi$ is locally bounded, the switching rate $(t,z) \mapsto \lambda(t,z)$ is a continuous function, and the jump kernel $Q$ is such that $Q(t, x, \cdot)$ is a probability distribution. In particular these assumptions imply that

$$\sup_{t \in [0,T], |z| \leqslant \rho} \int_{\mathbb{R}^d} (1 \wedge |z-y|^2) \lambda(t,z) Q(t,z,\mathrm{d}y) \leqslant \sup_{t \in [0,T], |z| \leqslant \rho} \lambda(t,z) < \infty \quad \text{for all } \rho \geqslant 0.$$

Then, Conforti and Léonard [2022, Theorem 5.7] requires a further integrability condition, which is satisfied when

$$\int_{[0,T] \times \mathbb{R}^d \times \mathbb{R}^d} (1 \wedge |z-y|^2) \mu_0 P_t(\mathrm{d}z) \lambda(t,z) Q(t,z,\mathrm{d}y) < \infty.$$

It is then sufficient to have that

$$\int_0^{\mathrm{T}_f} \mathbb{E}[\lambda(t, Z_t)] \mathrm{d}t < \infty \tag{16}$$

Finally, Conforti and Léonard [2022, Theorem 5.7] requires some technical assumptions which we now discuss. Introduce the class of functions that are twice continuously differentiable and compactly supported, denoted by $\mathcal{C}_c^2(\mathbb{R}^d)$, and for $f \in \mathcal{C}_c^2(\mathbb{R}^d)$ consider the two following conditions:

$$\int_0^T \int_{\mathbb{R}^d} \mu_0 P_t(\mathrm{d}z) |\mathscr{L}_t f(z)| \mathrm{d}t < \infty, \tag{17}$$

$$\int_0^T \int_{\mathbb{R}^d} \mu_0 P_t(\mathrm{d}z) |\Gamma_t(f,g)(z)| \mathrm{d}t < \infty \qquad \text{for all } g \in \mathcal{C}_c^2(\mathbb{R}^d). \qquad (18)$$

We define $\mathcal{F} := \{ f \in \mathcal{C}_c^2(\mathsf{E}) : (17), (18) \text{ hold} \}$. We need to verify that $\mathcal{F} \equiv \mathcal{C}_c^2(\mathsf{E})$. Let us start by considering (17): we find

$$\int_0^T \mu_0 P_t(\mathrm{d}z) |\mathscr{L}_t f(z)| \mathrm{d}t$$

$$\leqslant \int_0^T \mu_0 P_t(\mathrm{d}z) \left( |\langle \Phi(t,z), \nabla f(z) \rangle| + \lambda(t,z) \int |u(y) - u(z)| Q(t,z,\mathrm{d}y) \right) \mathrm{d}t.$$

Since $f \in \mathcal{C}_c^2(\mathsf{E})$ we have that $|\langle \Phi(t,z), \nabla f(z) \rangle|$ is compactly supported and hence integrable, while the second term is finite assuming $\int_0^T \mathbb{E}[\lambda(t,Z_t)] \mathrm{d}t < \infty$. Under the latter assumption, (18) can be easily verified.

## A.5 Proof of Proposition 2

Let us denote the initial condition of the forward PDMP by $\mu_0 = \mu_0^X \otimes \mu_0^V$. First of all, notice that, for a PDMP with position-velocity decomposition and homogeneous jump kernel, the flux equation (3) becomes

$$\mu_0 P_{\tilde{t}}(\mathrm{d}y, \mathrm{d}w) \overleftarrow{\lambda}(t,(y,w)) \overleftarrow{Q}(t,(y,w),(\mathrm{d}x,\mathrm{d}v)) = \mu_0 P_{\tilde{t}}(\mathrm{d}x,\mathrm{d}v) \lambda(\tilde{t},(x,v)) Q((x,v),(\mathrm{d}y,\mathrm{d}w))$$

where $\tilde{t} = \mathrm{T}_f - t$. Moreover, since the jump kernel leaves the position vector unchanged we obtain that this is equivalent to

$$\mu_0 P_{\tilde{t}}(\mathrm{d}w|y) \overleftarrow{\lambda}(t,(y,w)) \overleftarrow{Q}(t,(y,w),(\mathrm{d}x,\mathrm{d}v)) = \mu_0 P_{\tilde{t}}(\mathrm{d}v|y) \lambda(\tilde{t},(y,v)) Q((x,v),(\mathrm{d}y,\mathrm{d}w)),$$

where $\mu_0 P_t(\mathrm{d}w|y)$ is the conditional law of the velocity vector given the position vector at time $t$ with initial distribution $\mu_0$.

Suppose first that $Q((y,w),(\mathrm{d}x,\mathrm{d}v)) = \delta_y(\mathrm{d}x) \delta_{\mathscr{R}_y w}(\mathrm{d}v)$ for an involution $\mathscr{R}_y$. Then we find

$$\mu_0 P_{\tilde{t}}(\mathrm{d}w|y) \overleftarrow{\lambda}(t,(y,w)) \overleftarrow{Q}(t,(y,w),(\mathrm{d}x,\mathrm{d}v)) = \mu_0 P_{\tilde{t}}(\mathrm{d}\mathscr{R}_y w|y) \lambda(\tilde{t},(y,\mathscr{R}_y w)) \delta_y(\mathrm{d}x) \delta_{\mathscr{R}_y w}(\mathrm{d}v)$$

where we used that $\delta_{\mathscr{R}_y w}(\mathrm{d}v) = \delta_{\mathscr{R}_y v}(\mathrm{d}w)$ since $\mathscr{R}_y$ is an involution. Under our assumptions we have

$$\mu_0 P_{\tilde{t}}(\mathrm{d}\mathscr{R}_y w|y) = p_{\tilde{t}}(\mathscr{R}_y w|y) \mu_{\mathrm{ref}}^V(\mathrm{d}w), \quad \mu_0 P_{\tilde{t}}(\mathrm{d}w|y) = p_{\tilde{t}}(w|y) \mu_{\mathrm{ref}}^V(\mathrm{d}w),$$

since we assumed $\mu_{\mathrm{ref}}^V(\mathrm{d}w) = \mu_{\mathrm{ref}}^V(\mathrm{d}\mathscr{R}_y w)$. Hence we find for any $(y,w) \in \mathbb{R}^{2d}$ such that $p_{\tilde{t}}(w|y) > 0$

$$\overleftarrow{\lambda}(t,(y,w)) \overleftarrow{Q}(t,(y,w),(\mathrm{d}x,\mathrm{d}v)) = \frac{p_{\tilde{t}}(\mathscr{R}_y w|y)}{p_{\tilde{t}}(w|y)} \lambda(\tilde{t},(y,\mathscr{R}_y w)) \delta_y(\mathrm{d}x) \delta_{\mathscr{R}_y w}(\mathrm{d}v).$$

This can only be satisfied if

$$\overleftarrow{\lambda}(t,(y,w)) = \frac{p_{\tilde{t}}(\mathscr{R}_y w|y)}{p_{\tilde{t}}(w|y)} \lambda(\tilde{t},(y,\mathscr{R}_y w)), \quad Q(t,(y,w),(\mathrm{d}x,\mathrm{d}v)) = \delta_y(\mathrm{d}x) \delta_{\mathscr{R}_y w}(\mathrm{d}v).$$

Consider now the second case, that is $Q((y,w),(\mathrm{d}x,\mathrm{d}v)) = \delta_y(\mathrm{d}x) \nu(\mathrm{d}v|y)$ and $\lambda(t,(y,w)) = \lambda(t,y)$. The flux equation (3) can be rewritten as

$$\mu_0 P_{\tilde{t}}(\mathrm{d}w|y) \overleftarrow{\lambda}(t,(y,w)) \overleftarrow{Q}(t,(y,w),(\mathrm{d}x,\mathrm{d}v)) = \mu_0 P_{\tilde{t}}(\mathrm{d}v|y) \lambda(\tilde{t},y) \delta_y(\mathrm{d}x) \nu(\mathrm{d}w|y)$$

Under our assumptions we have $\nu(\mathrm{d}w|y) = (\mathrm{d}\nu/\mathrm{d}\mu_{\mathrm{ref}}^V)(w|y) \mu_{\mathrm{ref}}^V(\mathrm{d}w)$ and $\mu_0 P_{\tilde{t}}(\mathrm{d}w|y) = p_{\tilde{t}}(w|y) \mu_{\mathrm{ref}}^V(\mathrm{d}w)$ for some measure $\mu_{\mathrm{ref}}^V$. Hence for any $(y,w) \in \mathbb{R}^{2d}$ such that $p_{\tilde{t}}(w|y) > 0$ we obtain

$$\overleftarrow{\lambda}(t,(y,w)) \overleftarrow{Q}(t,(y,w),(\mathrm{d}x,\mathrm{d}v)) = \frac{(\mathrm{d}\nu/\mathrm{d}\mu_{\mathrm{ref}}^V)(w|y)}{p_{\tilde{t}}(w|y)} \lambda(\tilde{t},y) p_{\tilde{t}}(\mathrm{d}v|y) \delta_y(\mathrm{d}x).$$

This is satisfied when

$$\overleftarrow{\lambda}(t,(y,w)) = \frac{(\mathrm{d}\nu/\mathrm{d}\mu_{\mathrm{ref}}^V)(w|y)}{p_{\tilde{t}}(w|y)} \lambda(\tilde{t},y), \quad \overleftarrow{Q}(t,(y,w),(\mathrm{d}x,\mathrm{d}v)) = \mu_0 P_{\tilde{t}}(\mathrm{d}v|y) \delta_y(\mathrm{d}x).$$

## A.6 Extension of Proposition 2 to multiple jump types

Proposition 2 considers PDMPs with one type of jump, while here we discuss the case of characteristics of the form (14), which is e.g. the case of ZZP and BPS. In this setting we can assume the backward jump rate and kernel have a similar structure, that is

$$\overleftarrow{\lambda}(t,z) = \sum_{i=1}^{\ell} \overleftarrow{\lambda}_i(t,z) \,, \quad \overleftarrow{Q}(t,z,\mathrm{d}z') = \sum_{i=1}^{\ell} \frac{\overleftarrow{\lambda}_i(t,z)}{\overleftarrow{\lambda}(t,z)} \overleftarrow{Q}_i(t,z,\mathrm{d}z') \,,$$

in which case the balance condition (3) can be rewritten as

$$\mu_0 P_{\mathrm{T}_f-t}(\mathrm{d}y) \sum_{i=1}^{\ell} \overleftarrow{\lambda}_i(t,y) \overleftarrow{Q}_i(t,y,\mathrm{d}z) = \mu_0 P_{\mathrm{T}_f-t}(\mathrm{d}z) \sum_{i=1}^{\ell} \lambda_i(\mathrm{T}_f-t,z) Q_i(\mathrm{T}_f-t,z,\mathrm{d}y) \,.$$

It is then enough that

$$\mu_0 P_{\mathrm{T}_f-t}(\mathrm{d}y) \overleftarrow{\lambda}_i(t,y) \overleftarrow{Q}_i(t,y,\mathrm{d}z) = \mu_0 P_{\mathrm{T}_f-t}(\mathrm{d}z) \lambda_i(\mathrm{T}_f-t,z) Q_i(\mathrm{T}_f-t,z,\mathrm{d}y)$$

holds for all $i \in \{1,\ldots,\ell\}$. It follows that it is sufficient to apply Proposition 2 to each pair $(\lambda_i, Q_i)$ to obtain $(\overleftarrow{\lambda}_i, \overleftarrow{Q}_i)$ such that (3) holds.

## A.7 Time reversals of ZZP, BPS, and RHMC

In this section we give rigorous statements regarding time reversals of ZZP, BPS, and RHMC. For all samplers we rely on Proposition 2 and hence we focus on verifying its assumptions. In the cases of ZZP and RHMC we assume the technical condition **H**4 since proving it rigorously is out of the scope of the present paper. We remark that this can be proved with techniques as in Durmus et al. [2021], which show **H**4 in the case of BPS.

**Proposition 4 (Time reversal of ZZP)** *Consider a ZZP $(X_t, V_t)_{t\in[0,\mathrm{T}_f]}$ with initial distribution $\mu_\star \otimes \nu$, where $\nu = \mathrm{Unif}(\{\pm 1\}^d)$ and invariant distribution $\pi \otimes \nu$, where $\pi$ has potential $\psi$ satisfying **H**1. Assume that **H**4 holds and that $\int \mu_\star(\mathrm{d}x)|\partial_i\psi(x)| < \infty$ for all $i = 1,\ldots,d$. Then the time reversal of the ZZP has vector field*

$$\overleftarrow{\Phi}^{\mathrm{Z}}(x,v) = (-v,0)^T$$

*and jump rates and kernels are given for all $(y,w) \in \mathbb{R}^{2d}$ such that $P_{\mathrm{T}_f-t}(w|y) > 0$ by*

$$\overleftarrow{\lambda}_i^{\mathrm{Z}}(t,(y,w)) = \frac{p_{\mathrm{T}_f-t}(\mathscr{R}_i^{\mathrm{Z}}w|y)}{p_{\mathrm{T}_f-t}(w|y)} \lambda_i^{\mathrm{Z}}(y,\mathscr{R}_i^{\mathrm{Z}}w), \quad \overleftarrow{Q}_i^{\mathrm{Z}}((y,w),(\mathrm{d}x,v)) = \delta_{(y,\mathscr{R}_i^{\mathrm{Z}}w)}(\mathrm{d}x,v)$$

*for $i = 1,\ldots,d$.*

**Proof** We verify the conditions of Proposition 2 corresponding to deterministic transitions and rely on Appendix A.6 to apply the proposition to each pair $(\lambda_i^{\mathrm{Z}}, Q_i^{\mathrm{Z}})$. First notice the vector field $\Phi(x,v) = (v,0)^T$ is clearly locally bounded and $(t,x) \mapsto \lambda(x,v)$ is continuous since $\psi$ is continuously differentiable. Moreover, the ZZP can be shown to be non-explosive applying Durmus et al. [2021, Proposition 9]. Then, we need to verify (16). First, observe that $\mathbb{E}[\lambda_i(X_t, V_t)] \leqslant \mathbb{E}[|\partial_i\psi(X_t)|]$. Then

$$\mathbb{E}[|\partial_i\psi(X_t)|] = \mathbb{E}\left[\left|\partial_i\psi(X_0) + \int_0^1 \langle X_t - X_0, \nabla\partial_i\psi(X_0 + s(X_t - X_0))\rangle\mathrm{d}s\right|\right]$$

$$\leqslant \mathbb{E}[|\partial_i\psi(X_0)|] + \mathbb{E}\left[\int_0^1 |\langle X_t - X_0, \nabla^2\psi(X_0 + s(X_t - X_0))\mathrm{e}_i\rangle|\mathrm{d}s\right]$$

where $\mathrm{e}_i$ is the $i$-th vector of the canonical basis. Notice that $|X_t - X_0| \leqslant t\sqrt{d}$. Thus we find

$$\mathbb{E}[|\partial_i\psi(X_t)|] \leqslant \mathbb{E}[|\partial_i\psi(X_0)|] + t\sqrt{d} \sup_{x\in\mathbb{R}^d}\|\nabla^2\psi(x)\|$$

and therefore

$$\int_0^{T_f} \mathbb{E}[\lambda(X_t, V_t)]\mathrm{d}t \leqslant T_f \sum_{i=1}^d \left( \mathbb{E}|\partial_i \psi(X_0)| + \frac{T_f}{2}\sqrt{d} \sup_{x \in \mathbb{R}^d} \|\nabla^2 \psi(x)\| \right).$$

Since $\mathbb{E}_{\mu_\star}|\partial_i \psi(X)| < \infty$ and because we are assuming **H**1, we obtain (16). Finally, notice that $P_t(\mathrm{d}v|x)$ is absolutely continuous with respect to the counting measure on $\{1, -1\}^d$, which is clearly invariant with respect to $\mathscr{R}_i^{\mathrm{Z}}$. $\qquad\square$

**Proposition 5 (Time reversal of BPS)** *Consider a BPS* $(X_t, V_t)_{t \in [0, T_f]}$ *with initial distribution* $\mu_\star \otimes \nu$, *where* $\nu = \mathrm{Unif}(\mathsf{S}^{d-1})$, *and invariant distribution* $\pi \otimes \nu$, *where* $\pi$ *has potential* $\psi$ *satisfying* **H**1. *Assume that* $\mathbb{E}_{\mu_\star}[|\nabla \psi(X)|] < \infty$. *Then there exists a density* $p_t(w|y) := {}^{\mathrm{d}(\mu_0 P_t)(\mathrm{d}w|y)}/_{\nu(\mathrm{d}w)}$. *Moreover, the time reversal of the BPS has vector field*

$$\overleftarrow{\Phi}^{\mathrm{B}}(x, v) = (-v, 0)^T,$$

*while the jump rates and kernels are given for all* $t, y, w \in [0, T_f] \times \mathbb{R}^{2d}$ *such that* $p_{T_f - t}(w|y) > 0$ *by*

$$\overleftarrow{\lambda}_1^{\mathrm{B}}(t, (y, w)) = \frac{p_{\tilde{t}}(\mathscr{R}_y^{\mathrm{B}}w|y)}{p_{\tilde{t}}(w|y)}\lambda_1^{\mathrm{B}}(y, \mathscr{R}_y^{\mathrm{B}}w), \quad \overleftarrow{Q}_1^{\mathrm{B}}((y, w), (\mathrm{d}x, \mathrm{d}v)) = \delta_{(y, \mathscr{R}_y^{\mathrm{B}}w)}(\mathrm{d}x, \mathrm{d}v),$$

$$\overleftarrow{\lambda}_2^{\mathrm{B}}(t, (y, w)) = \lambda_r \frac{1}{p_{T_f - t}(w|y)}, \quad \overleftarrow{Q}_2^{\mathrm{B}}(t, (y, w), (\mathrm{d}x, \mathrm{d}v)) = \mu_0 P_{T_f - t}(\mathrm{d}v|y)\delta_y(\mathrm{d}x)\mathrm{d}v, \quad (19)$$

*where* $\tilde{t} = T_f - t$.

**Remark 3** *Under the assumption that* $\nu$ *is the uniform distribution on the sphere, it is natural to take* $\mu_{\mathrm{ref}}^{\mathrm{V}} = \nu$, *which gives that* ${}^{\mathrm{d}\nu}/_{\mathrm{d}\mu_{\mathrm{ref}}^{\mathrm{V}}} = 1$ *and hence the backward refreshment rate is as in* (19). *When* $\nu$ *is the d-dimensional Gaussian distribution, the natural choice is to let* $\mu_{\mathrm{ref}}^{\mathrm{V}}$ *be the Lebesgue measure and hence we obtain a rate as given in* (5).

**Proof** We verify the general conditions of Proposition 2, then focusing on the deterministic jumps and the refreshments relying on Appendix A.6. The BPS was shown to be non-explosive for any initial distribution in Durmus et al. [2021, Proposition 10]. Since $\lambda(t, (x, v)) = \lambda(x, v) = \langle v, \nabla \psi(x) \rangle_+$, with a similar reasoning of the proof of Proposition 4 we have

$$\mathbb{E}[\lambda(X_t, V_t)] = \mathbb{E}\left[ \left( \langle V_t, \nabla \psi(X_0) \rangle + \int_0^1 \langle V_t, \nabla^2 \psi(X_0 + s(X_t - X_0))(X_t - X_0) \rangle \mathrm{d}s \right)_+ \right].$$

Taking advantage of $|V_t| = 1$ we have $|X_t - X_0| \leqslant t$ and thus we find

$$\mathbb{E}[\lambda(X_t, V_t)] \leqslant \mathbb{E}\left[ |\nabla \psi(X_0)| + \int_0^1 |\nabla^2 \psi(X_0 + s(X_t - X_0))(X_t - X_0)|\mathrm{d}s \right]$$
$$\leqslant \mathbb{E}\left[ |\nabla \psi(X_0)| \right] + t \sup_{x \in \mathbb{R}^d} \|\nabla^2 \psi(x)\|.$$

This is sufficient to obtain (16) since $\mathbb{E}[|\nabla \psi(X_0)|] < \infty$ and we assume **H**1. Moreover, **H**4 holds by Durmus et al. [2021, Proposition 23]. Finally notice that $P_t(\mathrm{d}v|x)$ is absolutely continuous with respect to $\mu_{\mathrm{ref}}^{\mathrm{V}} = \mathrm{Unif}(\mathsf{S}^{d-1})$, which satisfies $\mu_{\mathrm{ref}}^{\mathrm{B}}(\mathscr{R}^{\mathrm{B}}(x)v) = \mu_{\mathrm{ref}}^{\mathrm{B}}(v)$ for all $x, v \in \mathbb{R}^d \times \mathsf{S}^{d-1}$. All the required assumptions in Proposition 2 are thus satisfied. $\qquad\square$

**Proposition 6 (Time reversal of RHMC)** *Consider a RHMC* $(X_t, V_t)_{t \in [0, T_f]}$ *with initial distribution* $\mu_\star \otimes \nu$, *where* $\nu$ *is the d-dimensional standard normal distribution, and invariant distribution* $\pi \otimes \nu$, *where* $\pi$ *has potential* $\psi \in \mathcal{C}^1(\mathbb{R}^d)$. *Suppose that* **H**4 *holds and that for any* $y \in \mathbb{R}^d$, $P_t(\mathrm{d}w|y)$ *is absolutely continuous with respect to Lebesgue measure, with density* $p_t(w|y)$. *Then the time reversal of the RHMC has vector field*

$$\overleftarrow{\Phi}^{\mathrm{H}}(x, v) = (-v, \nabla \psi(x))^T,$$

*while the jump rates and kernels are given for all* $(y, w) \in \mathbb{R}^{2d}$ *such that* $p_{T_f - t}(w|y) > 0$ *by*

$$\overleftarrow{\lambda}_2^{\mathrm{H}}(t, (y, w)) = \lambda_r \frac{\nu(w)}{p_{T_f - t}(w|y)}, \quad \overleftarrow{Q}_2^{\mathrm{H}}(t, (y, w), (\mathrm{d}x, \mathrm{d}v)) = p_{T_f - t}(v|y)\delta_y(\mathrm{d}x)\mathrm{d}v.$$

**Proof** First of all, RHMC is non-explosive by Durmus et al. [2021, Proposition 8]. Then $\Phi$ is locally bounded and (16) is trivially satisfied. Finally, we can take $\mu_{\text{ref}}^{\text{V}}$ to be the Lebesgue measure.
$\square$

## B Density ratio matching

### B.1 Ratio matching with Bregman divergences

We now describe a general approach to approximate ratios of densities based on the minimisation of Bregman divergences [Sugiyama et al., 2011], which as we discuss is closely connected to the loss of Hyvärinen [2007].

For a differentiable, strictly convex function $f$ we define the Bregman divergence $B_f(r,s) := f(r) - f(s) - f'(s)(r - s)$. Given two time-dependent probability density functions on $\mathbb{R}^{2d}$, $p, q$, we wish to approximate their ratio $r(x, v, t) = p_t(x,v)/q_t(x,v)$ for $t \in [0, T_f]$ with a parametric function $s_\theta : \mathbb{R}^d \times \mathbb{R}^d \times [0, T_f] \to \mathbb{R}_+$ by solving the minimisation problem

$$\min_\theta \int_0^{T_f} \omega(t) \, \mathbb{E}\Big[ B_f(r(X_t, V_t, t), s_\theta(X_t, V_t, t)) \Big] \mathrm{d}t,$$

where the expectation is with respect to the joint density $q_t(x, v)$, that is $(X_t, V_t) \sim q_t$, while $\omega$ is a probability density function for the time variable. Well studied choices of the function $f$ include e.g. $f(r) = r \log r - r$, that is related to a KL divergence, or $f(r) = (r-1)^2$, related to the square loss, or $f(r) = r \log r - (1 + r) \log(1 + r)$, which corresponding to solving a logistic regression task. Ignoring terms that do not depend on $\theta$ we can rewrite the minimisation as

$$\min_\theta \int_0^{T_f} \omega(t) \left( \mathbb{E}_{p_t}\big[ f'(s_\theta(X_t, V_t, t)) s_\theta(X_t, V_t, t) - f(s_\theta(X_t, V_t, t)) \big] - \mathbb{E}_{q_t}\big[ f'(s_\theta(X_t, V_t, t)) \big] \right) \mathrm{d}t.$$

Notably this is independent of the true density ratio and thus it is a formulation with similar spirit to *implicit score matching*. Naturally, in practice the loss can be approximated empirically with a Monte Carlo average.

### B.2 Details and proofs regarding Hyvärinen's ratio matching

#### B.2.1 Connection to Bregman divergences

In the next statement, we show that the loss $\ell_\mathrm{I}$ defined in (6), or equivalently its explicit counterpart $\ell_\mathrm{E}$ (see Proposition 3), can be put in the framework of Bregman divergences.

**Corollary 1** *Recall* $\mathbf{G}(r) = (1 + r)^{-1}$ *and let* $f(r) = (r-1)^2/2$. *The task* $\min_\theta \ell_\mathrm{E}(\theta)$ *is equivalent to*

$$\min_\theta \sum_{i=1}^d \mathbb{E}_{p_t}\Big[ B_f(\mathbf{G}(s_i^\theta(X_t, V_t, t)), \mathbf{G}(r_i(X_t, V_t, t)))$$
$$+ B_f(\mathbf{G}(s_i^\theta(X_t, \mathscr{R}_i^{\mathrm{Z}} V_t, t)), \mathbf{G}(r_i(X_t, \mathscr{R}_i^{\mathrm{Z}} V_t, t))) \Big]$$

**Proof** The result follows by straightforward computations. $\square$

#### B.2.2 Proof of Proposition 3

The proof follows the same lines as Hyvärinen [2007, Theorem 1]. We find

$$\ell_\mathrm{E}(\theta) = C + \int_0^{T_f} \omega(t) \sum_{i=1}^d \mathbb{E}_{p_t}\Big[ \mathbf{G}^2(s_i^\theta(X_t, V_t, t)) + \mathbf{G}^2(s_i^\theta(X_t, \mathscr{R}_i^{\mathrm{Z}} V_t, t))$$
$$- 2\mathbf{G}(r_i(X_t, V_t, t))\mathbf{G}(s_i^\theta(X_t, V_t, t)) - 2\mathbf{G}(s_i^\theta(X_t, \mathscr{R}_i^{\mathrm{Z}} V_t, t))\mathbf{G}(r_i(X_t, \mathscr{R}_i^{\mathrm{Z}} V_t, t)) \Big] \mathrm{d}t,$$

---

**Algorithm 1:** Pseudo-code for the simulation of a homogeneous PDMP

---

**Input** : Time horizon T, initial condition $(x, v)$.
Set $t = 0$, $(X_0, V_0) = (x, v)$;
**while** $t < $ T **do**
    draw $E \sim \text{Exp}(1)$;
    compute the next event time, that is $\tau \in \mathbb{R}_+$ such that $\int_0^\tau \lambda(\varphi_u(X_t, V_t))\mathrm{d}u = E$ ;
    **if** $t + \tau > T$ **then**
       | set $Z_T = \varphi_{T-t}(Z_t)$;
    **else**
       | simulate $Z_{t+\tau} \sim Q(\varphi_s(Z_t), \cdot)$;
    **end**
    set $t = t + \tau$;
**end**

---

where $C$ is a constant independent of $\theta$. Then plugging in the expression of $\mathbf{G}$ we can rewrite the last term as

$$
\mathbb{E}_{p_t}\Big[\mathbf{G}(s_i^\theta(X_t, \mathscr{R}_i^Z V_t, t))\mathbf{G}(r_i(X_t, \mathscr{R}_i^Z V_t, t))\Big]
$$

$$
= \int \sum_{v \in \{\pm 1\}^d} p_t(x, v)\mathbf{G}(s_i^\theta(x, \mathscr{R}_i^Z v, t)) \frac{p_t(x, \mathscr{R}_i^Z v)}{p_t(x, v) + p_t(x, \mathscr{R}_i^Z v)} \mathrm{d}x
$$

$$
= \mathbb{E}_{p_t}\Big[\mathbf{G}(s_i^\theta(X_t, V_t, t)) \frac{p_t(X_t, \mathscr{R}_i^Z V_t)}{p_t(X_t, V_t) + p_t(X_t, \mathscr{R}_i^Z V_t)}\Big].
$$

Therefore we find

$$
\ell_{\mathrm{E}}(\theta) = C + \int_0^{T_f} \omega(t) \sum_{i=1}^d \mathbb{E}_{p_t}\Big[\mathbf{G}^2(s_i^\theta(X_t, V_t, t)) + \mathbf{G}^2(s_i^\theta(X_t, \mathscr{R}_i^Z V_t, t))
$$

$$
- \frac{2\mathbf{G}(s_i^\theta(X_t, V_t, t))\, p_t(X_t, V_t)}{p_t(X_t, V_t) + p_t(X_t, \mathscr{R}_i^Z V_t)} - \frac{2\mathbf{G}(s_i^\theta(X_t, V_t, t))\, p_t(X_t, \mathscr{R}_i^Z V_t)}{p_t(X_t, V_t) + p_t(X_t, \mathscr{R}_i^Z V_t)}\Big]\mathrm{d}t
$$

$$
= C + \int_0^{T_f} \omega(t) \sum_{i=1}^d \mathbb{E}_{p_t}\Big[\mathbf{G}^2(s_i^\theta(X_t, V_t, t)) + \mathbf{G}^2(s_i^\theta(X_t, \mathscr{R}_i^Z V_t, t)) - 2\mathbf{G}(s_i^\theta(X_t, V_t, t))\Big]\mathrm{d}t.
$$

## C   Simulation of forward and backward PDMPs

### C.1   Simulation of the forward PDMPs

The forward PDMPs that we consider can all be simulated exactly by solving the integral (13), at least when the limiting distribution is the multivariate standard normal. This is possible because for each process we can easily simulate the random event times as well as their deterministic dynamics. The general procedure for the simulation of a time-homogeneous PDMP up to a fixed time horizon T is given in Algorithm 1. In the remainder of the section we give additional details on the simulation of each process.

**RHMC:**   The case of RHMC is trivial, since the random events have the exponential distribution with constant parameter $\lambda_r$ and the deterministic dynamics are given by $x_t = x_0 \cos(t) + v_0 \sin(t)$ and $v_t = -x_0 \sin(t) + v_0 \cos(t)$, where $(x_0, v_0)$ is the initial condition. Hence all the steps in Algorithm 1 can be performed and the state $(X_T, V_T)$ can be easily obtained.

**ZZP:**   Notice the event rates of ZZP are of the form $\lambda_i(x, v) = (v_i x_i)_+$ when the stationary distribution is the standard normal. In this case we find that each coordinate of the ZZP is independent, that is the evolution of $((X_t)_i, (V_t)_i)$ is not affected by $((X_t)_j, (V_t)_j)$ for $i, j = 1, \ldots, d$ with $i \neq j$. Therefore we can simulate each coordinate of the ZZP in parallel following the procedure in Algorithm 1. Let us illustrate how one can obtain the next event time $\tau$ of the $i$-th coordinate when the

process is at $(x_i, v_i)$. We have that $\tau$ solves $\int_0^\tau (v_i x_i + u)_+ \mathrm{d}u - E = 0$ for $E \sim \mathrm{Exp}(1)$. This gives the following quadratic equation for $\tau$:

$$\frac{\tau^2}{2} + v_i x_i \tau - v_i x_i (-v_i x_i)_+ - \frac{1}{2}(-v_i x_i)_+^2 - E = 0,$$

which has solution

$$\tau = -v_i x_i + \sqrt{(v_i x_i)_+^2 + 2E}.$$

**BPS:**  In the case of BPS one has to simulate a proposal for both the next reflection event, $\tau_b$ and for the next refreshment event, $\tau_r$, then accepting the smallest of the two as event time. Obtaining the proposal for the next refreshment time is straightforward since the rate is constant. The proposal for the following reflection time can be obtained similarly to the case of ZZP, but noticing that in this case the event rate is $\lambda(x, v) = \langle v, x \rangle_+$. Then $\tau_b$ solves $\int_0^{\tau_b} (\langle v, x \rangle + |v|^2 u)_+ \mathrm{d}u - E = 0$ for $E \sim \mathrm{Exp}(1)$. Noticing that it must be $\tau_b > \langle v,x \rangle / |v|^2$, this gives the following quadratic equation for $\tau_b$:

$$\frac{|v|^2}{2}\tau_b^2 + \langle v, x \rangle \tau_b + \frac{1}{2}(-\langle v, x \rangle)_+^2 - E = 0,$$

which has solution

$$\tau_b = \frac{-\langle v, x \rangle + \sqrt{\langle v, x \rangle_+^2 + 2|v|^2 E}}{|v|^2}.$$

### C.2   Simulation of time reversed PDMPs with splitting schemes

Here we discuss the splitting schemes we use to discretise the backward PDMPs. For further details on this class of approximations we refer the reader to Bertazzi et al. [2023].

**RHMC:**  We have already discussed the splitting scheme DJD for RHMC in Section 2.4, and we give the pseudo-code in Algorithm 2.

---

**Algorithm 2:** Splitting scheme DJD for the time reversed RHMC

---

Initialise either from $(\overline{X}_0, \overline{V}_0) \sim \pi \otimes \nu$ or $(\overline{X}_0, \overline{V}_0) \sim \pi(\mathrm{d}x)p_{\theta*}(\mathrm{d}v|x, \mathrm{T}_f)$;

**for** $n = 0, \dots, N-1$ **do**

$\quad (\widetilde{X}, \widetilde{V}) = \varphi^{\mathrm{H}}_{-\delta_{n+1/2}}(\overline{X}_{t_n}, \overline{V}_{t_n})$ ;

$\quad \widetilde{t} = \mathrm{T}_f - t_n - \frac{\delta_{n+1}}{2}$ ;

$\quad$ Estimate ratio: $\overline{s} = \nu(\widetilde{V}) / p_{\theta*}(\widetilde{V}| \widetilde{X}, \widetilde{t})$ ;

$\quad$ Draw proposal $\tau_{n+1} \sim \mathrm{Exp}(\overline{s}\lambda_r)$ ;

$\quad$ **if** $\tau_{n+1} \leqslant \delta_{n+1}$ **then**

$\quad\quad$ Draw $\widetilde{V} \sim p_{\theta*}(\cdot | \widetilde{X}, \widetilde{t})$;

$\quad$ **end**

$\quad (\overline{X}_{t_{n+1}}, \overline{V}_{t_{n+1}}) = \varphi^{\mathrm{H}}_{-\delta_{n+1/2}}(\widetilde{X}, \widetilde{V})$;

**end**

---

**ZZP:**  For ZZP we apply the splitting scheme DJD discussed in Section 2.4, with the only difference that we allow multiple velocity flips during the jump step similarly to Bertazzi et al. [2023]. Algorithm 3 gives a pseudo-code.

**BPS:**  In the case of BPS, we follow the recommendations of Bertazzi et al. [2023] and adapt their splitting scheme RDBDR, where R stands for refreshments, D for deterministic motion, and B for bounces. We give a pseudo-code in Algorithm 4. We remark that an alternative is to use the scheme DJD for BPS, simulating reflections and refreshments in the J part of the splitting. This choice has the advantage of reducing the number of model evaluations.

---
**Algorithm 3:** Splitting scheme DJD for the time reversed ZZP
---
Initialise $(\overline{X}_0, \overline{V}_0) \sim \pi \otimes \nu$;

**for** $n = 0, \ldots, N-1$ **do**
$\quad \widetilde{X} = \overline{X}_{t_n} - \frac{\delta_{n+1}}{2} \overline{V}_{t_n}$ ;
$\quad \widetilde{V} = \overline{V}_{t_n}$ ;
$\quad \widetilde{t} = T_f - t_n - \frac{\delta_{n+1}}{2}$ ;
$\quad$ Estimate density ratios: $s^{\theta^*}(\widetilde{X}, \widetilde{V}, \widetilde{t})$ ;
$\quad$ **for** $i = 1 \ldots, d$ **do**
$\quad\quad |\quad$ With probability $(1 - \exp(-\delta_{n+1} s_i^{\theta^*}(\widetilde{X}, \widetilde{V}, \widetilde{t}) \lambda_i(\widetilde{X}, \mathscr{R}_i^Z \widetilde{V})))$ set $\widetilde{V} = \mathscr{R}_i^Z \widetilde{V}$ ;
$\quad$ **end**
$\quad \overline{X}_{t_{n+1}} = \widetilde{X} - \frac{\delta_{n+1}}{2} \widetilde{V}$ ;
$\quad \overline{V}_{t_{n+1}} = \widehat{V}$ ;
**end**

---

---
**Algorithm 4:** Splitting scheme RDBDR for the time reversed BPS
---
Initialise either from $(\overline{X}_0, \overline{V}_0) \sim \pi \otimes \nu$ or $(\overline{X}_0, \overline{V}_0) \sim \pi(\mathrm{d}x) p_{\theta^*}(\,\cdot\,|x, T_f)$ ;

**for** $n = 0, \ldots, N-1$ **do**
$\quad \widetilde{V} = \overline{V}_{t_n}$ ;
$\quad$ Estimate density ratio: $\overline{s}_2 = {\nu(\widetilde{V})}/{p_{\theta^*}(\widetilde{V}|\overline{X}_{t_n}, T_f-t_n)}$ ;
$\quad$ With probability $(1 - \exp(-\lambda_r \overline{s}_2 \frac{\delta_{n+1}}{2}))$ draw $\widetilde{V} \sim p_{\theta^*}(\,\cdot\,|\overline{X}_{t_n}, T_f - t_n)$ ;
$\quad \widetilde{X} = \overline{X}_{t_n} - \frac{\delta_{n+1}}{2} \overline{V}_{t_n}$ ;
$\quad \widetilde{t} = T_f - t_n - \frac{\delta_{n+1}}{2}$ ;
$\quad$ Estimate density ratio: $\overline{s}_1 = {p_{\theta^*}(\mathscr{R}_{\widetilde{X}}^B \widetilde{V}|\widetilde{X}, \widetilde{t})}/{p_{\theta^*}(\widetilde{V}|\widetilde{X}, \widetilde{t})}$ ;
$\quad$ With probability $(1 - \exp(-\delta_{n+1} \overline{s}_1 \lambda_1(\widetilde{X}, \mathscr{R}_{\widetilde{X}}^B \widetilde{V})))$ set $\widetilde{V} = \mathscr{R}_{\widetilde{X}}^B \widetilde{V}$ ;
$\quad \overline{X}_{t_{n+1}} = \widetilde{X} - \frac{\delta_{n+1}}{2} \widetilde{V}$ ;
$\quad$ Estimate density ratio: $\overline{s}_2 = {\nu(\widetilde{V})}/{p_{\theta^*}(\widetilde{V}|\overline{X}_{t_{n+1}}, T_f-t_{n+1})}$ ;
$\quad$ With probability $(1 - \exp(-\lambda_r \overline{s}_2 \frac{\delta_{n+1}}{2}))$ draw $\widetilde{V} \sim p_{\theta^*}(\,\cdot\,|\overline{X}_{t_n}, T_f - t_{n+1})$ ;
$\quad \overline{V}_{t_{n+1}} = \widetilde{V}$ ;
**end**

---

## D    Training the generative models

In this section, we present the algorithmic procedures used to train our generative models, and outline computational differences with the popular framework of denoising diffusion models. In Table 3 we list the backward rates and kernels of the time reversal associated with each forward PDMP introduced in Section 2.1. In Appendix D.1 we give the procedure used for ZZP, while in Appendix D.2 we focus on RHMC and BPS, which can be trained with the same approach. In Appendix D.3 we compare the training phase of PDMP-based and diffusion-based generative models.

### D.1    Fitting the ZZP-based generative model

In the case of ZZP, we only need to approximate the jump rates $\overleftarrow{\lambda}_i^Z$ of the backward process, since we have access to the true backward kernel. To this end, we introduce a class of functions $\{s^\theta : \mathbb{R}^d \times \{-1, 1\}^d \times [0, T_f] \to \mathbb{R}_+^d \,:\, \theta \in \Theta\}$ for some parameter set $\Theta \subset \mathbb{R}^{d_\theta}$ such that for any $i \in \{1, \ldots, d\}$, the $i$-th component of $s^\theta(y, w, t)$, denoted by $s_i^\theta(y, w, t)$, is an approximation of

$$r_i^Z(y, w, t) = \frac{p_{T_f - t}(\mathscr{R}_i^Z w|y)}{p_{T_f - t}(w|y)} \ .$$

| Process | Backward Rates | Backward Kernels |
|---|---|---|
| ZZP | for $i = 1, \ldots, d$ 
 $\overleftarrow{\lambda}_i^{\mathrm{Z}}(t, (y, w)) = \dfrac{p_{\mathrm{T}_f - t}(\mathscr{R}_i^{\mathrm{Z}} w \mid y)}{p_{\mathrm{T}_f - t}(w \mid y)} \lambda_i^{\mathrm{Z}}(y, \mathscr{R}_i^{\mathrm{Z}} w)$ | for $i = 1, \ldots, d$ 
 $\overleftarrow{Q}_i^{\mathrm{Z}}((y, w), (\mathrm{d}x, \mathrm{d}v)) = \delta_{(y, \mathscr{R}_i^{\mathrm{Z}} w)}(\mathrm{d}x, \mathrm{d}v)$ |
| RHMC | $\overleftarrow{\lambda}^{\mathrm{H}}(t, (y, w)) = \lambda_r \dfrac{\nu(w)}{p_{\mathrm{T}_f - t}(w \mid y)}$ | $\overleftarrow{Q}^{\mathrm{H}}(t, (y, w), (\mathrm{d}x, \mathrm{d}v)) = p_{\mathrm{T}_f - t}(v \mid y)\, \delta_y(\mathrm{d}x)\, \mathrm{d}v$ |
| BPS | $\overleftarrow{\lambda}_1^{\mathrm{B}}(t, (y, w)) = \dfrac{p_{\mathrm{T}_f - t}(\mathscr{R}_y^{\mathrm{B}} w \mid y)}{p_{\mathrm{T}_f - t}(w \mid y)} \lambda_1^{\mathrm{B}}(y, \mathscr{R}_y^{\mathrm{B}} w)$ 
 $\overleftarrow{\lambda}_2^{\mathrm{B}}(t, (y, w)) = \lambda_r \dfrac{\nu(w)}{p_{\mathrm{T}_f - t}(w \mid y)}$ | $\overleftarrow{Q}_1^{\mathrm{B}}((y, w), (\mathrm{d}x, \mathrm{d}v)) = \delta_{(y, \mathscr{R}_y^{\mathrm{B}} w)}(\mathrm{d}x, \mathrm{d}v)$ 
 $\overleftarrow{Q}_2^{\mathrm{B}}(t, (y, w), (\mathrm{d}x, \mathrm{d}v)) = p_{\mathrm{T}_f - t}(v \mid y)\, \delta_y(\mathrm{d}x)\, \mathrm{d}v$ |

Table 3: Time Reversal Characteristics of ZZP, RHMC, BPS.

We learn $s^\theta$ by minimising the empirical counterpart of the implicit ratio matching loss $\ell_{\mathrm{I}}$ given in (7). We define such empirical loss $\hat{\ell}_{\mathrm{I}}$ as the function $\hat{\ell}_{\mathrm{I}} : \theta \mapsto \hat{L}_{\mathrm{I}}(s_\theta, (X_{\tau^n}^n, V_{\tau^n}^n, \tau^n)_{n=1}^N)$ for

$$
\hat{L}_{\mathrm{I}}(s_\theta, (X_{\tau^n}^n, V_{\tau^n}^n, \tau^n)_{n=1}^N)
$$
$$
= \frac{1}{N} \sum_{n=1}^N \sum_{i=1}^d \left( \mathbf{G}^2(s_i^\theta(X_{\tau^n}^n, V_{\tau^n}^n, \tau^n)) + \mathbf{G}^2(s_i^\theta(X_{\tau^n}^n, \mathscr{R}_i^{\mathrm{Z}} V_{\tau^n}^n, \tau^n)) - 2\mathbf{G}(s_i^\theta(X_{\tau^n}^n, V_{\tau^n}^n, \tau^n)) \right) ,
$$

(20)

where $\{\tau^n\}_{n=1}^N$ are i.i.d. samples from $\omega$, independent of $\{(X_t^n, V_t^n)_{t \geqslant 0}\}_{n=1}^N$, which are $N$ i.i.d. realisations of the ZZP respectively starting at the $n$-th training data point $X_0^n$ with velocity $V_0^n$, where $\{V_0^n\}_{n=1}^N$ are i.i.d. observations of $\mathrm{Unif}(\{-1, 1\}^d)$. Algorithm 5 shows the pseudo code for our training algorithm. We remark that the simulation of the forward ZZP follows the guidelines explained in Appendix C.1.

---

**Algorithm 5:** Training loop for ZZP-based generative models

---

**Input:** Time distribution $\omega$ on $[0, \mathrm{T}_f]$, model $s^\theta$
**while** $\theta$ *has not converged* **do**
    Get random data batch $\{X_0^n\}_{n=1}^B$;
    Sample $\{\tau^n\}_{n=1}^B \sim \omega^{\otimes B}$;
    Sample $\{V_0^n\}_{n=1}^B \sim \mathrm{Unif}(\{\pm 1\}^d)^{\otimes B}$;
    **for** $n = 1$ *to* $B$ **do**
        Simulate $(X_{\tau^n}^n, V_{\tau^n}^n)$ by running a ZZP from $(X_0^n, V_0^n)$;
    Compute loss $\hat{\ell}_{\mathrm{I}}(\theta) \leftarrow \hat{L}_{\mathrm{I}}(s^\theta, (X_{\tau^n}^n, V_{\tau^n}^n, \tau^n)_{n=1}^B)$ ;
    Perform optimization step on $\hat{\ell}_{\mathrm{I}}(\theta)$;
**Output:** trained model $s^{\theta^*}$

---

**A computationally efficient model for the ZZP.** The computation of $\hat{L}_{\mathrm{I}}$ in each iteration of Algorithm 5 requires evaluating $d + 1$ times the model $s^\theta$ for each data-point. Indeed, the loss function (7) requires evaluating $s^\theta$ for each component flip of the velocity vector. Hence the computational cost of the algorithm scales linearly in the data dimension $d$.

In order to overcome this computational burden we build an alternative model leveraging the following observation:

$$
p_t(v_i | x, v_{-i}) \approx p_t(v_i | x), \tag{21}
$$

where $v_{-i}$ denotes the vector containing all components of $v$ other than the $i$-th. The approximation in (21) is motivated by the fact that we expect the components of the velocity to be nearly independent conditional on the position vector. This is because the position of the process holds most of the information regarding the velocity of each coordinate. Under (21) we find that a good approximation for the ratio $r_i^{\mathrm{Z}}(x, v, t)$ is given by

$$
\bar{r}_i^{\mathrm{Z}}(x, v_i, t) := \frac{p_t(-v_i | x)}{p_t(v_i | x)} .
$$

Hence it is reasonable to estimate the simplified ratios $\bar{r}_i^Z$ rather than the true ratios $r_i^Z(x, v, t)$. In particular this can be achieved with a model $s^\theta$ that requires the current position and time, and only the $i$-th velocity component rather than the full vector $v$.

Following the reasoning above we build an alternative model which takes as input the position vector and the time variable and outputs a $2d$-dimensional vector, that is $\{s^\theta : \mathbb{R}^d \times [0, \mathrm{T}_f] \to \mathbb{R}_+^{2d} : \theta \in \Theta\}$. We introduce the following notation for the output of the neural network $s^\theta$:

$$s^\theta : (x, t) \mapsto \left(s_+^\theta(x, t), \ s_-^\theta(x, t)\right) \in \mathbb{R}_+^{2d}, \tag{22}$$

where $s_+^\theta$ and $s_-^\theta$ are both vectors in $\mathbb{R}^d$ and denote the two, $d$-dimensional blocks in the output of $s^\theta$. The model will be trained in such a way that $s_{+,i}^\theta(x, t)$, that is the $i$-th component of the vector $s_+^\theta(x, t)$, approximates $\bar{r}_i^Z(x, +1, t)$, i.e. in the case $v_i = +1$. Similarly, we estimate $\bar{r}_i^Z(x, -1, t)$ with $s_{-,i}^\theta(x, t)$.

Let us now describe how we can train this model in such a way that the computational cost remains constant in the dimensionality of the data. For any $w \in \{1, -1\}^d$, we introduce the projection operator $\Pi_w^i$ defined for any $w \in \{\pm 1\}^d$, $i \in \{1, \ldots, d\}$, $y \in \mathbb{R}^d$, $t \in [0, \mathrm{T}_f]$ as

$$\Pi_w^i s^\theta(y, t) = s_{\mathrm{sign}(w_i),i}^\theta(y, t), \tag{23}$$

where we have $\mathrm{sign}(w_i) = +$ when $w_i = +1$ and $\mathrm{sign}(w_i) = -$ when $w_i = -1$. In words, $\Pi_w^i s^\theta(y, t)$ selects either $s_+^\theta$ or $s_-^\theta$ in (22) based on $\mathrm{sign}(w_i)$ and returns the estimate of the $i$-th ratio at $(y, t)$ corresponding to the velocity $w_i$. Using the operator $\Pi_w^i$ we can re-formulate the loss (7) as

$$\tilde{\ell}_\mathrm{I}(\theta) = \int_0^{\mathrm{T}_f} \mathrm{d}t\, \omega(t) \sum_{i=1}^d \mathbb{E}\left[\mathbf{G}^2(\Pi_{V_t}^i s^\theta(X_t, t)) + \mathbf{G}^2(\Pi_{-V_t}^i s^\theta(X_t, t)) - 2\mathbf{G}(\Pi_{V_t}^i s^\theta(X_t, t))\right].$$

The associated empirical loss is $\hat{\ell}_\mathrm{I} : \theta \mapsto \hat{L}_\mathrm{I}^{\mathrm{Simple}}(s^\theta, (X_{\tau^n}^n, V_{\tau^n}^n, \tau^n)_{n=1}^N)$ were

$$\begin{aligned}
&\hat{L}_\mathrm{I}^{\mathrm{Simple}}(s^\theta, (x^n, v^n, t^n)_{n=1}^N) = \\
&\frac{1}{N} \sum_{n=1}^N \sum_{i=1}^d \left(\mathbf{G}^2(\Pi_{v^n}^i s^\theta(x^n, t^n)) + \mathbf{G}^2(\Pi_{-v^n}^i s^\theta(x^n, t^n)) - 2\mathbf{G}(\Pi_{v^n}^i s^\theta(x^n, t^n))\right),
\end{aligned} \tag{24}$$

where $\{\tau^n\}_{n=1}^N$ and $\{(X_t^n, V_t^n)_{t \geqslant 0}\}_{n=1}^N$ are obtained as in (20). This simplified model can then be trained using the same procedure shown in Algorithm 5, but where the loss above is used instead of the loss (20). The computational cost for the training of this model is clearly constant in the data dimension $d$, since only a single evaluation of the model $s^\theta$ is needed for each data-point.

### D.2 Fitting the RHMC and BPS-based generative models

In the case of RHMC and BPS both the backward rate and backward jump kernel are characterised by $p_t(v|x)$. We introduce a parametric family of conditional probability distributions $\{p_\theta : \theta \in \Theta\}$ of the form $(x, v, t) \mapsto p_\theta(v|x, t)$, where $\Theta \subset \mathbb{R}^{d_\theta}$, which we model with normalising flows (NFs) [Papamakarios et al., 2021], as it permits both obtaining an estimate of the density, at a given state and time, and also to generate samples according to this distribution. To train our model, we rely on the maximum likelihood population loss method, as derived in (8), and use its empirical counterpart (9). The final BPS and RHMC training algorithm is given in Algorithm 6. The simulation of the forward BPS and RHMC follows the methods listed in Appendix C.1.

### D.3 Computational comparison with diffusion models

We provide a short comparison of the computational complexity of our PDMP generative methods with diffusion models, as each generative method admits the same design: a fixed forward process $\{X_t\}_{0 \leqslant t \leqslant \mathrm{T}_f}$ ran from time 0 to $\mathrm{T}_f$, with $X_{\mathrm{T}_f}$ approximately distributed as a standard Gaussian $\mathcal{N}(0, \mathrm{I}_d)$ (using the Variance-Preserving process in the case of diffusion models [Song et al., 2021]), a corresponding backward process $\{\overleftarrow{X}_t\}_{0 \leqslant t \leqslant \mathrm{T}_f}$, and a generative process $\{\overleftarrow{X}_t^\theta\}_{0 \leqslant t \leqslant \mathrm{T}_f}$ being an approximation to the true backward process, initialized from $\overleftarrow{X}_0^\theta \sim \mathcal{N}(0, \mathrm{I}_d)$.

**Algorithm 6:** Training loop for RHMC and BPS based generative models

**Input:** Time distribution $\omega$ on $[0, \mathrm{T}_f]$, model $p_\theta$
**while** $\theta$ *has not converged* **do**
     Get random data batch $\{X_0^n\}_{n=1}^B$;
     Sample $\{\tau^n\}_{n=1}^B \sim \omega^{\otimes B}$;
     Sample $\{V_0^n\}_{n=1}^B \sim \mathcal{N}(0, \mathrm{I}_d)^{\otimes B}$;
     **for** $n = 1$ *to* $B$ **do**
         Simulate $(X_{\tau^n}^n, V_{\tau^n}^n)$ running a RHMC/BPS from $(X_0^n, V_0^n)$;
     $L^\theta \leftarrow -\frac{1}{B} \sum_{n=1}^B \log p_\theta(V_{\tau^n}^n | X_{\tau^n}, \tau^n)$;
     Perform optimisation step on $L^\theta$;
**Output:** trained model $p_{\theta*}$

Using conventional notations for diffusion models [Song et al., 2021] we denote by $(\bar\alpha_t)_{0 \leqslant t \leqslant \mathrm{T}_f}$ the variance-preserving noise schedule, such that the distribution of the forward process at any time $t$ is given by

$$X_t \stackrel{d}{=} \sqrt{\bar\alpha_t} X_0 + \sqrt{1 - \bar\alpha_t} G_t , \tag{25}$$

where $G_t \sim \mathcal{N}(0, \mathrm{I}_d)$. The generative process $\{\overleftarrow{X}_t^\theta\}_{0 \leqslant t \leqslant \mathrm{T}_f}$ is typically defined by a denoiser neural network $\epsilon_\theta, \theta \in \mathbb{R}^{d_\theta}$, trained with the denoiser loss

$$\ell_{\text{diffusion}}^\theta = \mathbb{E}\left[\left\|\epsilon_\theta(X_t, t) - \frac{X_t - \sqrt{\bar\alpha_t} X_0}{\sqrt{1 - \bar\alpha_t}}\right\|_2^2\right] ,$$

where $t \sim \omega$ is the time parameter. We display the training algorithm for diffusion models in Algorithm 8. For the sake of comparison we give in Algorithm 7 the general procedure used for PDMP-based generative models.

| **Algorithm 7:** Training loop for PDMP-based generative models | **Algorithm 8:** Training loop for diffusion-based generative models |
|---|---|
| **Input:** Time distribution $\omega$ on $[0, \mathrm{T}_f]$, model $s^\theta$ 
 **while** $\theta$ *has not converged* **do** 
   Get random data batch $\{X_0^n\}_{n=1}^B$; 
   Sample $\{\tau^n\}_{n=1}^B \sim \omega^{\otimes B}$; 
   Sample $\{V_0^n\}_{n=1}^B \sim \mathrm{Unif}(\{\pm 1\}^d)^{\otimes B}$; 
   **for** $n = 1$ *to* $B$ **do** 
     Simulate $(X_{\tau^n}^n, V_{\tau^n}^n)$ by running 
     PDMP from $(X_0^n, V_0^n)$; 
   Compute loss $L^\theta$ ; 

   Perform optimization step on $L^\theta$; 
 **Output:** trained model $s^{\theta^*}$ | **Input:** Time distribution $\omega$ on $[0, \mathrm{T}_f]$, noise schedule $\{\bar\alpha_t\}_{t=0}^{\mathrm{T}_f}$, model $\epsilon^\theta$ 
 **while** $\theta$ *has not converged* **do** 
   Get random data batch $\{X_0^n\}_{n=1}^B$; 
   Sample $\{\tau^n\}_{n=1}^B \sim \omega^{\otimes B}$; 
   Sample $\{\epsilon^n\}_{n=1}^B \sim \mathcal{N}(0, \mathrm{I}_d)^{\otimes B}$; 
   **for** $n = 1$ *to* $B$ **do** 
     Compute 
     $X_{\tau^n}^n = \sqrt{\bar\alpha_{\tau^n}} X_0^n + \sqrt{1 - \bar\alpha_{\tau^n}} \epsilon^n$; 
   Compute loss 
   $L^\theta \leftarrow \frac{1}{B} \sum_{n=1}^B \left\|\epsilon^\theta(X_{\tau^n}^n, \tau^n) - \epsilon^n\right\|^2$; 
   Perform optimization step on $L^\theta$; 
 **Output:** trained model $\epsilon^{\theta^*}$ |

**Computational differences in training the models**    We compare the training algorithms:

- The simulation of the forward process $\{X_t\}_{0 \leqslant t \leqslant \mathrm{T}_f}$ is more efficient in the case of diffusion models, since the distribution of $X_t$ given $X_0$ is explicitly characterizable as given in (25), whereas in the case of PDMPs the complexity scales with the expected number of random jumps in $[0, \mathrm{T}_f]$.

- Computing the loss function has the same computational cost (number of evaluations of the network relative to the dimension of the data) for PDMP and diffusion models (assuming we are using the simplified model for ZZP, as introduced in Appendix D.1).

**Computational differences in the generative processes**   Since we are using the splitting scheme for our PDMPs, simulating the backward processes admits a computational complexity growing linearly with the chosen number of backward steps $N$, alike diffusion models [Song et al., 2021]. The latter models require only a single network inference per backward step. However, as measured in Table 2, each backward step is costlier in the case of our PDMP samplers. We provide an explanation for each sampler:

- **ZZP** Using the simplified model introduced in Appendix D.1 and Algorithm 3, only one network inference is required per backward step, to approximate the backward rate. However, this approach requires a slight modification to the neural network architecture, involving a doubled channel output and adjustments for the element selection mechanisms (projection operator (23)), resulting in a higher cost per step than diffusion models.

- **RHMC** Using Algorithm 2, one likelihood computation from the normalizing flow is needed, to approximate the backward rate. Additionally, at most one random variable must be drawn from the normalizing flow, which approximates the backward kernel.

- **BPS** Using Algorithm 4, three likelihood computations are required from the normalizing flow, to approximate backward rates. Moreover, at most two random variables need to be sampled from the normalizing flow, which approximates the backward kernel. The final cost per step depends on the chosen normalizing flow architecture, but is higher for BPS than for all other samplers.

It should be noted that, in our experiments, each generative PDMP model requires less backward steps than the diffusion model, leading to a smaller overall computational time for equal generation quality, as showed in Figure 3.

## E   Discussion and proof for Theorem 1

### E.1   Discussion on H2

In this section we discuss **H**2 in the case of ZZP, BPS, and RHMC. For all three of these samplers, existing theory shows convergence of the form

$$\|\delta_{(x,v)}P_t - \pi \otimes \nu\|_V \leqslant C'e^{-\gamma t}V(x,v), \tag{26}$$

where $V : \mathbb{R}^{2d} \to [1, \infty)$ is a positive function and $\|\mu\|_V := \sup_{|g| \leqslant V} |\mu(g)|$ is the $V$-norm. When the initial condition of the process is $\mu_\star \otimes \nu$, we obtain the bound

$$\|\mu_\star \otimes \nu P_t - \pi \otimes \nu\|_V \leqslant C'e^{-\gamma t}\mu_\star \otimes \nu(V),$$

which translates to a bound in TV distance, since we assume $V \geqslant 1$. Conditions on $\pi$ ensuring (26) can be found for ZZP in Bierkens et al. [2019b], for BPS in Deligiannidis et al. [2019], Durmus et al. [2020], and for RHMC in Bou-Rabee and Sanz-Serna [2017]. Observe that we can set the constant $C$ in **H**2 to $C = C'\mu_\star \otimes \nu(V)$. Clearly, $C$ is finite whenever $\mu_\star \otimes \nu(V) < \infty$. Since $V$ is such that $\lim_{|z| \to \infty} V(z) = +\infty$, showing $C$ is finite requires suitable tail conditions on the initial distribution $\mu_\star \otimes \nu$. Inspecting the results of the papers mentioned above, one can verify that $\mu_\star \otimes \nu(V) < \infty$ when $\pi$ is a multivariate standard Gaussian distribution as long as: (i) the tails of $\mu_\star$ are at least as light as those of $\pi$ for ZZP and BPS, (ii) $\mu_\star$ has finite second moments for RHMC.

### E.2   Proof of Theorem 1

First notice that
$$\|\mu_\star - \mathcal{L}(\overline{X}_{\mathrm{T}_f})\|_{\mathrm{TV}} \leqslant \|\mu_\star \otimes \nu - \mathcal{L}(\overline{X}_{\mathrm{T}_f}, \overline{V}_{\mathrm{T}_f})\|_{\mathrm{TV}}, \tag{27}$$
hence we focus on bounding the right hand side. Under our assumptions, the forward PDMP $(X_t, V_t)_{t \in [0, \mathrm{T}_f]}$ admits a time reversal that is a PDMP $(\overleftarrow{X}_t, \overleftarrow{V}_t)_{t \in [0, \mathrm{T}_f]}$ with characteristics $(\overleftarrow{\Phi}, \overleftarrow{\lambda}, \overleftarrow{Q})$ satisfying the conditions in Proposition 1. Therefore, it holds $\mu_\star \otimes \nu = \mathcal{L}(\overleftarrow{X}_{\mathrm{T}_f}, \overleftarrow{V}_{\mathrm{T}_f})$ and so (27) can be written as

$$\|\mu_\star - \mathcal{L}(\overline{X}_{\mathrm{T}_f})\|_{\mathrm{TV}} \leqslant \|\mathcal{L}(\overleftarrow{X}_{\mathrm{T}_f}, \overleftarrow{V}_{\mathrm{T}_f}) - \mathcal{L}(\overline{X}_{\mathrm{T}_f}, \overline{V}_{\mathrm{T}_f})\|_{\mathrm{TV}},$$

We introduce the intermediate PDMP $(\widetilde{X}_t, \widetilde{V}_t)_{t\in[0,\mathrm{T}_f]}$ with initial distribution $\mathcal{L}(X_{\mathrm{T}_f}, V_{\mathrm{T}_f})$ and characteristics $(\overleftarrow{\Phi}, \overline{\lambda}, \overline{Q})$. In particular, $(\widetilde{X}_t, \widetilde{V}_t)_{t\in[0,\mathrm{T}_f]}$ has the same characteristics as $(\overline{X}_t, \overline{V}_t)_{t\in[0,\mathrm{T}_f]}$, but different initial condition By the triangle inequality for the TV distance we find

$$\|\mu_\star - \mathcal{L}(\overline{X}_{\mathrm{T}_f})\|_{\mathrm{TV}} \leqslant \|\mathcal{L}(\overleftarrow{X}_{\mathrm{T}_f}, \overleftarrow{V}_{\mathrm{T}_f}) - \mathcal{L}(\widetilde{X}_{\mathrm{T}_f}, \widetilde{V}_{\mathrm{T}_f})\|_{\mathrm{TV}} + \|\mathcal{L}(\widetilde{X}_T, \widetilde{V}_{\mathrm{T}_f}) - \mathcal{L}(\overline{X}_{\mathrm{T}_f}, \overline{V}_{\mathrm{T}_f})\|_{\mathrm{TV}}.$$

Applying the data processing inequality to the second term, we find the bound

$$\|\mu_\star - \mathcal{L}(\overline{X}_{\mathrm{T}_f})\|_{\mathrm{TV}} \leqslant \|\mathcal{L}(\overleftarrow{X}_{\mathrm{T}_f}, \overleftarrow{V}_{\mathrm{T}_f}) - \mathcal{L}(\widetilde{X}_{\mathrm{T}_f}, \widetilde{V}_{\mathrm{T}_f})\|_{\mathrm{TV}} + \|\mathcal{L}(X_{\mathrm{T}_f}, V_{\mathrm{T}_f}) - \pi \otimes \nu\|_{\mathrm{TV}}. \quad (28)$$

The second term in (28) can be bounded applying **H**2, hence it is left to bound the first term. We introduce the Markov semigroups $P_t, \overleftarrow{P}_t, \overline{P}_t : \mathbb{R}_+ \times \mathbb{R}^{2d} \times \mathcal{B}(\mathbb{R}^{2d}) \to [0,1]$ defined respectively as $P_t((x,v), \cdot) := \mathbb{P}_{(x,v)}((X_t, V_t) \in \cdot)$, $\overleftarrow{P}_t((x,v), \cdot) := \mathbb{P}_{(x,v)}((\overleftarrow{X}_t, \overleftarrow{V}_t) \in \cdot)$, and $\widetilde{P}_t((x,v), \cdot) := \mathbb{P}_{(x,v)}((\widetilde{X}_t, \widetilde{V}_t) \in \cdot)$. Recall that for any probability distribution $\eta$ on $(\mathbb{R}^{2d}, \mathcal{B}(\mathbb{R}^{2d}))$, $\eta P_t(\cdot) = \int_{\mathbb{R}^{2d}} \eta(\mathrm{d}x, \mathrm{d}v) P_t((x,v), \cdot)$, and similarly for $\eta \widetilde{P}_t(\cdot)$ and $\eta \overleftarrow{P}_t(\cdot)$. Finally, to ease the notation we denote $Q_{\mathrm{T}_f} := \mathcal{L}(X_{\mathrm{T}_f}, V_{\mathrm{T}_f}) = (\mu_\star \otimes \nu) P_{\mathrm{T}_f}$. Then we can rewrite the first term in (28) as

$$\|\mathcal{L}(\overleftarrow{X}_{\mathrm{T}_f}, \overleftarrow{V}_{\mathrm{T}_f}) - \mathcal{L}(\widetilde{X}_{\mathrm{T}_f}, \widetilde{V}_{\mathrm{T}_f})\|_{\mathrm{TV}} = \|Q_{\mathrm{T}_f} \overleftarrow{P}_{\mathrm{T}_f} - Q_{\mathrm{T}_f} \widetilde{P}_{\mathrm{T}_f}\|_{\mathrm{TV}}$$
$$\leqslant \int Q_{\mathrm{T}_f}(\mathrm{d}x, \mathrm{d}v) \|\delta_{(x,v)} \overleftarrow{P}_{\mathrm{T}_f} - \delta_{(x,v)} \widetilde{P}_{\mathrm{T}_f}\|_{\mathrm{TV}}. \quad (29)$$

Therefore we wish to bound $\|\delta_{(x,v)} \overleftarrow{P}_{\mathrm{T}_f} - \delta_{(x,v)} \widetilde{P}_{\mathrm{T}_f}\|_{\mathrm{TV}}$. A bound for the TV distance between two PDMPs with same initial condition and deterministic motion, but different jump rate and kernel was obtained in Durmus et al. [2021, Theorem 11] using the coupling inequality

$$\|\delta_{(x,v)} \overleftarrow{P}_{\mathrm{T}_f} - \delta_{(x,v)} \widetilde{P}_{\mathrm{T}_f}\|_{\mathrm{TV}} \leqslant 2\mathbb{P}_{(x,v)}\left((\overleftarrow{X}_{\mathrm{T}_f}, \overleftarrow{V}_{\mathrm{T}_f}) \neq (\widetilde{X}_{\mathrm{T}_f}, \widetilde{V}_{\mathrm{T}_f})\right),$$

and then bounding the right hand side. Following the proof of Durmus et al. [2021, Theorem 11] we have that a synchronous coupling of the two PDMPs satisfies

$$\mathbb{P}_{(x,v)}\left((\overleftarrow{X}_{\mathrm{T}_f}, \overleftarrow{V}_{\mathrm{T}_f}) \neq (\widetilde{X}_{\mathrm{T}_f}, \widetilde{V}_{\mathrm{T}_f})\right) \leqslant 2\mathbb{E}_{(x,v)}\left[1 - \exp\left(-\int_0^{\mathrm{T}_f} g_t(\overleftarrow{X}_t, \overleftarrow{V}_t)\mathrm{d}t\right)\right],$$

where

$$g_t(x,v) = \frac{1}{2}\left(\overleftarrow{\lambda}(t, (x,v)) \wedge \overline{\lambda}(t, (x,v))\right)\left\|\overleftarrow{Q}(t, (x,v), \cdot) - \overline{Q}(t, (x,v), \cdot)\right\|_{\mathrm{TV}}$$
$$+ \left|\overleftarrow{\lambda}(t, (x,v)) - \overline{\lambda}(t, (x,v))\right|.$$

Since $\mathcal{L}(\overleftarrow{X}_t, \overleftarrow{V}_t) = \mathcal{L}(X_{\mathrm{T}_f-t}, V_{\mathrm{T}_f-t})$ for $t \in [0, \mathrm{T}_f]$, we can rewrite this bound as

$$\mathbb{P}_{(x,v)}\left((\overleftarrow{X}_{\mathrm{T}_f}, \overleftarrow{V}_{\mathrm{T}_f}) \neq (\widetilde{X}_{\mathrm{T}_f}, \widetilde{V}_{\mathrm{T}_f})\right) \leqslant 2\mathbb{E}_{(x,v)}\left[1 - \exp\left(-\int_0^{\mathrm{T}_f} g_{\mathrm{T}_f-t}(X_t, V_t)\mathrm{d}t\right)\right].$$

Plugging this bound in (29) we obtain

$$\|\mathcal{L}(\overleftarrow{X}_{\mathrm{T}_f}, \overleftarrow{V}_{\mathrm{T}_f}) - \mathcal{L}(\widetilde{X}_{\mathrm{T}_f}, \widetilde{V}_{\mathrm{T}_f})\|_{\mathrm{TV}} \leqslant 2\mathbb{E}\left[1 - \exp\left(-\int_0^{\mathrm{T}_f} g_{\mathrm{T}_f-t}(X_t, V_t)\mathrm{d}t\right)\right].$$

This concludes the proof.

### E.3 Application to the ZZP

Here we give the details on the bound (12), which considers the case of ZZP. First, we upper bound the function $g_t$ defined in (11). We focus on the first term in (11), that is

$$g_t^1(x,v) = \frac{(\overleftarrow{\lambda}^Z \wedge \bar{\lambda}^Z)(t, (x,v))}{2}\|\overleftarrow{Q}^Z(t, (x,v), \cdot) - \bar{Q}^Z(t, (x,v), \cdot)\|_{\mathrm{TV}}.$$

We find

$$\|\overleftarrow{Q}^{\mathsf{Z}}(t,(x,v),\cdot) - \bar{Q}^{\mathsf{Z}}(t,(x,v),\cdot)\|_{\mathrm{TV}} = \sup_A \left| \sum_{i=1}^d \mathbb{1}_{(x,\mathscr{R}_i^{\mathsf{Z}}v)\in A} \left( \frac{\overleftarrow{\lambda}_i^{\mathsf{Z}}(t,(x,v))}{\overleftarrow{\lambda}^{\mathsf{Z}}(t,(x,v))} - \frac{\bar{\lambda}_i^{\mathsf{Z}}(t,(x,v))}{\bar{\lambda}^{\mathsf{Z}}(t,(x,v))} \right) \right|$$

$$\leqslant \sup_A \left| \sum_{i=1}^d \mathbb{1}_{(x,\mathscr{R}_i^{\mathsf{Z}}v)\in A} \left( \frac{\overleftarrow{\lambda}_i^{\mathsf{Z}}(t,(x,v)) - \bar{\lambda}_i^{\mathsf{Z}}(t,(x,v))}{\overleftarrow{\lambda}^{\mathsf{Z}}(t,(x,v))} + \frac{\bar{\lambda}_i^{\mathsf{Z}}(t,(x,v))}{\overleftarrow{\lambda}^{\mathsf{Z}}(t,(x,v))} - \frac{\bar{\lambda}_i^{\mathsf{Z}}(t,(x,v))}{\bar{\lambda}^{\mathsf{Z}}(t,(x,v))} \right) \right|$$

$$\leqslant \left( \sum_{i=1}^d \frac{|\overleftarrow{\lambda}_i^{\mathsf{Z}}(t,(x,v)) - \bar{\lambda}_i^{\mathsf{Z}}(t,(x,v))|}{\overleftarrow{\lambda}^{\mathsf{Z}}(t,(x,v))} \right) + \frac{|\bar{\lambda}^{\mathsf{Z}}(t,(x,v)) - \overleftarrow{\lambda}^{\mathsf{Z}}(t,(x,v))|}{\overleftarrow{\lambda}^{\mathsf{Z}}(t,(x,v))}$$

In the last inequality we used that $\bar{\lambda}^{\mathsf{Z}}$ is non-negative. Therefore we find

$$g_t^1(x,v) \leqslant \frac{1}{2} \left( \sum_{i=1}^d |\overleftarrow{\lambda}_i^{\mathsf{Z}}(t,(x,v)) - \bar{\lambda}_i^{\mathsf{Z}}(t,(x,v))| \right) + \frac{1}{2} |\bar{\lambda}^{\mathsf{Z}}(t,(x,v)) - \overleftarrow{\lambda}^{\mathsf{Z}}(t,(x,v))|$$

$$\leqslant \sum_{i=1}^d |\overleftarrow{\lambda}_i^{\mathsf{Z}}(t,(x,v)) - \bar{\lambda}_i^{\mathsf{Z}}(t,(x,v))|.$$

Noticing that

$$|\overleftarrow{\lambda}_i^{\mathsf{Z}}(t,(x,v)) - \bar{\lambda}_i^{\mathsf{Z}}(t,(x,v))| = |r_i^{\mathsf{Z}}(x,v,t) - s_i^{\theta}(x,v,t)| \, \lambda_i^{\mathsf{Z}}((x,\mathscr{R}_i^{\mathsf{Z}}v).$$

we find

$$g_t(x,v) \leqslant 2 \sum_{i=1}^d |r_i^{\mathsf{Z}}(x,v,t) - s_i^{\theta}(x,v,t)| \, \lambda_i^{\mathsf{Z}}((x,\mathscr{R}_i^{\mathsf{Z}}v)$$

Finally, we use the inequality $1 - e^{-z} \leqslant z$, which holds for $z \geqslant 0$, to conclude that

$$\mathbb{E}\left[ 1 - \exp\left( -\int_0^{\mathrm{T}_f} g_{\mathrm{T}_f - t}(X_t, V_t)\mathrm{d}t \right) \right]$$

$$\leqslant 2 \sum_{i=1}^d \mathbb{E}\left[ \int_0^{\mathrm{T}_f} |r_i^{\mathsf{Z}}(X_t, V_t, \mathrm{T}_f - t) - s_i^{\theta}(X_t, V_t, \mathrm{T}_f - t)| \, \lambda_i^{\mathsf{Z}}(X_t, \mathscr{R}_i^{\mathsf{Z}}V_t)\mathrm{d}t \right].$$

# F  Experimental details

We run our experiments on 50 Cascade Lake Intel Xeon 5218 16 cores, 2.4GHz. Each experiment is ran on a single CPU and takes between 1 and 5 hours to complete, depending on the dataset and the sampler at hand.

For each forward PDMP, we take a time horizon $\mathrm{T}_f$ equal to 5, and set the refreshment rate $\lambda_r$ to 1. For training, we choose the uniform distribution $\mathrm{Uniform}([0, \mathrm{T}_f])$ as the time distribution $\omega$. For the simulation of backward PDMPs with splitting schemes, we use a quadratic schedule for the time steps, that is $(\delta_n)_{n\in\{1,\dots,N\}}$ given by $\delta_n = \mathrm{T}_f \times ((n/N)^2 - (n-1/N)^2)$.

For i-DDPM, we follow the design choices introduced in Nichol and Dhariwal [2021] and in particular we use the variance preserving (VP) process, the cosine noise schedule, and linear time steps.

## F.1  Continuation of Section 4

**2D datasets**  In our experiments we consider the five datasets displayed in Figure 5. The Gaussian grid consists of a mixture of nine Gaussian distribution with imbalanced mixture weights $\{.01, .02, .02, .05, .05, .1, .1, .15, .2, .3\}$. We load 100000 training samples for each dataset, and use 10000 test samples to compute the evaluation metrics. We use a batch size of 4096 and train our model for 25000 steps.

| refresh rate process | 0.0 | 0.1 | 0.5 | 1.0 | 2.0 | 5.0 | 10.0 |
|---|---|---|---|---|---|---|---|
| BPS | 0.286 | 0.069 | 0.048 | 0.052 | 0.045 | 0.048 | 0.072 |
| RHMC | 0.324 | 0.040 | 0.041 | 0.033 | 0.040 | 0.047 | 0.072 |
| ZZP | 0.040 | 0.045 | 0.040 | 0.036 | 0.038 | 0.035 | 0.057 |

Table 4: Mean of 2-Wasserstein $W_2 \downarrow$, on Gaussian grid dataset, averaged over 10 runs.

| time horizon process | 2 | 5 | 10 | 15 |
|---|---|---|---|---|
| BPS | 0.031 | 0.040 | 0.040 | 0.041 |
| HMC | 0.025 | 0.036 | 0.036 | 0.033 |
| ZZP | 0.031 | 0.033 | 0.030 | 0.046 |

Table 5: Mean 2-Wasserstein $W_2 \downarrow$ for different time horizon, averaged over 10 runs.

**Detailed setup** For ZZP and i-DDPM we use a neural network consisting of eight time-conditioned multi-layer perceptron (MLP) blocks with skip connections, each of which consisting of two fully connected layers of width 256. The time variable $t$ passes through two fully connected layers of size $1 \times 32$ and $32 \times 32$, and is fed to each time conditioned block, where it passes through an additional $32 \times 64$ fully connected layer before being added element-wise to the middle layer. The model size is 6.5 million parameters. For ZZP, we apply the `softplus` activation function $x \mapsto \frac{1}{\beta} \log(1 + \exp(\beta x))$ to the output of the network, with $\beta = 1$, to constrain it to be positive and stabilise behaviour for outputs close to 1.

In the case of RHMC and BPS, we use neural spline flows [Durkan et al., 2019] to model the conditional densities of the forward processes, as it shows good performance among available architectures. We leverage the implementation from the `zuko` package [Rozet et al., 2022]. We set the number of transforms to 8, the hidden depth of the network to 8 and the hidden width to 256. To condition on $x, t$, we feed them to three fully connected layers of size $d \times 8$, $8 \times 8$ and $8 \times 8$, where $d$ is either the dimension of $X_t$, or $d = 1$ in the case of the time variable. The resulting vectors are then concatenated and fed to the conditioning mechanism of `zuko`. The resulting model has 3.8 million parameters.

We take advantage of the approaches described in Section 2.3 to learn the characteristics of the backward processes.

All experiments are conducted using PyTorch [Paszke et al., 2019]. The optimiser is Adam [Kingma and Ba, 2015] with learning rate 5e-4 for all neural networks.

**Additional results** In Table 4 we show the accuracy in terms of the refreshment rate, while in Table 5 we show different choices of the time horizon. In both cases, we consider the Gaussian mixture data and we use the 2-Wasserstein metric to characterise the quality of the generated data. Figure 5 shows the generated data by the best model for each process.

### F.2 MNIST digits

We consider the task of generating handwritten digits training the ZZP on the MNIST dataset. We use the simplified loss function given in Equation (24) in Appendix D.1 and use the simplified model and loss. In Figure 6 we show promising results obtained with the design choices described previously in Appendix F.1, apart from the differences that follow. The optimiser is Adam [Kingma and Ba, 2015] with learning rate 2e-4. We use a U-Net following the implementation of Nichol and Dhariwal [2021], where the hidden layers are set to $[128, 256, 256, 256]$, where we fix the number of residual blocks to 2 at each level, and add self-attention block at resolution $16 \times 16$, with 4 heads. We duplicate the channel of the penultimate layer, and make each copy go through separate MLPs to obtain the two vectors $(s_+^\theta(\cdot, \cdot), s_-^\theta(\cdot, \cdot)) \in \mathbb{R}_+^{2d}$ (as in (22)). We use an exponential moving average with a rate of 0.99. At every layer, we use the `silu` activation function, while we apply

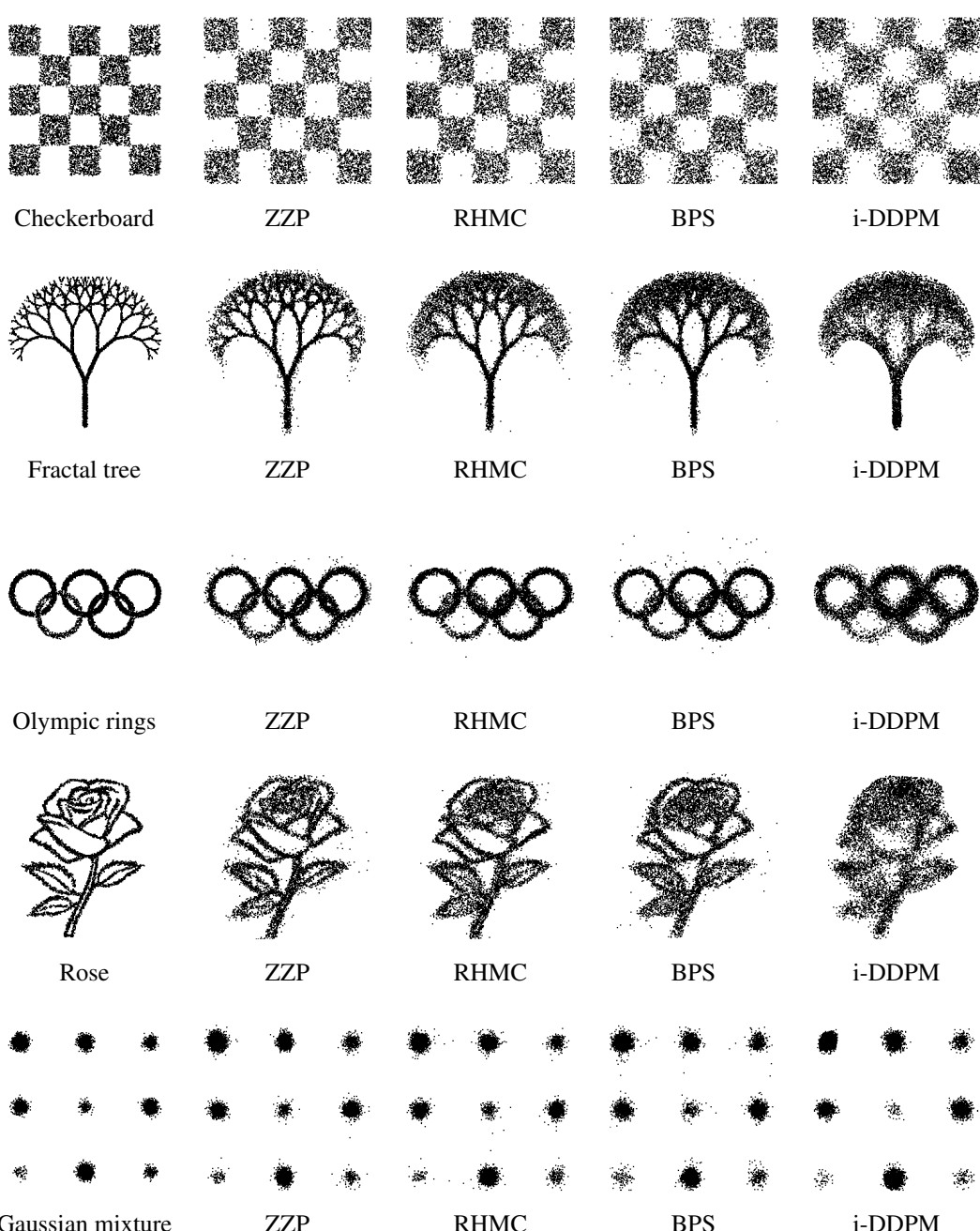

Figure 5: Generation for the various datasets.

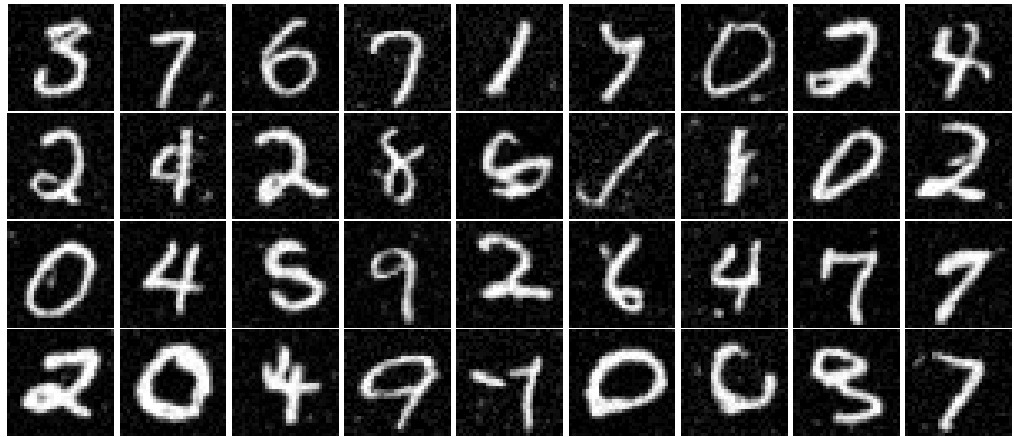

Figure 6: Generation for the ZZP trained on MNIST.

the `softplus` to the output of the network, with $\beta = 0.2$. We train the model for $40000$ steps with batch size $128$.

