# OpenReview forum: "Piecewise deterministic generative models"
_NeurIPS.cc/2024/Conference — NeurIPS 2024 poster_

### Official Review · Reviewer_jjQr · 2024-07-09

**Soundness:** 3
**Presentation:** 2
**Contribution:** 3
**Rating:** 7
**Confidence:** 4

**Summary:**

This paper proposes a new generative model on continuous random variables similar to diffusion-based generative models, but this method uses a different family of perturbation schemes. In particular, for the forward processes, which transform the clean data to a stationary distribution, the authors propose using piecewise deterministic Markov processes (PDMPs), which consist of a combination of discrete diffusions and ODEs. Specifically, the initial value, i.e., data, is transformed via simple ODEs, whose velocity term is often chosen to be a constant. However, the velocity term (also called vector fields) is updated at any random time with a constant rate $\lambda$. The occurrence of a new event is independent, and the time interval of the new event since the last one is exponentially distributed. Moreover, the new velocity will be assigned by user-chosen schemes; for example, one can flip the sign of the velocity, or another one can sample from the standard normal distribution. Thus, while the forward process transforms the data with a simple ODE, it randomly changes the ODEs, hence, the title of the paper.

In addition, the paper shows that under some mild assumptions, the time-reversed processes of the PDMPs are also PDMPs, and their transition probability distribution and update rate can be written in terms of the forward processes. Therefore, we can consider modeling such terms with parametric models similar to popular diffusion-based generative models.
The paper proposes the explicit ratio matching objective function, as in Equation 6, to train such terms.

Moreover, the authors suggest using three popular PDMPs for the forward process: Zig-Zag process (ZZP), bouncy particle sampler (BPS), and randomized Hamiltonian Monte Carlo (RHMC). Consequently, the paper shows the time-reversed PDMPs and potential parameterizations that fit each case.

Finally, the paper demonstrates the efficacy of the proposed method through a few toy experiments.

----------------------------------
Update the rating from 5 to 7 after the authors' rebuttal.

**Strengths:**

One of the paper's key contributions is the introduction of a novel generative model that uses PDMPs as the forward processes. Moreover, the authors propose a training method for the time-reversed PDMPs.

**Weaknesses:**

The proposed method is novel and interesting. However, the paper didn't characterize some potential drawbacks well. For example, due to the nature of PDMPs, the training of the proposed method inherits the problem of discrete diffusion models. In particular, the ratio matching should be done for each dimension independently, and this costs a lot for high-dimensional data compared to the denoising score matching and others for previous methods. This problem could be the reason that the current submission didn't include the experiments on popular high-dimensional datasets. In this regard, the paper suggests that Monte Carlo estimates can reduce the computational cost, but it trades off the variance of the learning signals, which would be critical for large-scale experiments.

I consider that a new method doesn't always need to be better than previous methods. Nevertheless, proper information about the proposed method in practice would be important for potential readers to evaluate the significance of the paper. Thus, some additional discussions related to this problem need to be added.

Moreover, given that many alternative choices exist other than the proposed method, in-depth discussions on the proposed method's merits would be appreciated for evaluating its importance.

In addition, I find that the presentation should be improved. For example, many statements contain multiple prepositional phrases, which are difficult to parse and understand.

**Questions:**

N/A

---

> ### Author Rebuttal · Authors · 2024-08-06
>
> Thank you for your thorough review and the time you have taken to provide valuable feedback on our paper. Below are our responses to your comments.
>
> * *Regarding the first two points mentioned in the Weaknesses*
>
>     We agree with the remarks and we will add clearer discussions on the limitations of our models in the next version of the paper. As the reviewer observed, the computational cost of the ratio matching loss in the case of ZZP grows linearly in d, which indeed makes the algorithm costly for higher dimensional datasets. As pointed out by the reviewer, subsampling over dimensions as suggested in lines 238-239 increases the variance of the unbiased estimator of the loss function. On the other hand, notice that the cost of the loss function associated with BPS and RHMC does not increase with the dimension, since the velocity vectors lie in a continuous space and the whole vector is updated at the same time, instead of flipping each component separately. Nevertheless, it is challenging to tune conditional normalising flows when the dimension increases, hence this constitutes the main limitation for BPS and RHMC.
>
> * *Moreover, given that many alternative choices exist other than the proposed method, in-depth discussions on the proposed method's merits would be appreciated for evaluating its importance.*
>
>     We agree with the reviewer that it is important to motivate our novel generative approach and we will add a paragraph to the introduction to improve this aspect. In general, we expect PDMPs to be successful at modelling data distributions that are supported on restricted domains, since it is very simple to adapt their deterministic dynamics to incorporate boundary terms (see e.g. [1]), especially if compared to diffusion processes. Similarly, data distributions that are a mixture of continuous density with point masses can be modelled adapting the ideas in [2], while extensions to data on Riemannian manifolds can be obtained leveraging flows which do not leave the manifold (see e.g. [3] for a PDMP on a sphere).
>
>     We also believe that further empirical studies will be beneficial to highlight the specific advantages of PDMPs over other methods. In the revised paper we plan to include comparisons with other generative models on the synthetic datasets we considered.
>
> * *In addition, I find that the presentation should be improved. For example, many statements contain multiple prepositional phrases, which are difficult to parse and understand.*
>
>     Concerning this remark by the reviewer, we interpret that it refers to sentences such as those in lines 136-138, or 316-318. We will rewrite these and similar sentences to make the presentation more streamlined and easy to read. In addition, we will modify the paper extensively in order to improve the general presentation, taking into consideration the feedback that we received by all reviewers. For example, we will improve the exposition of Section 2 by introducing intuitive explanations and visual aids such as trace plots of the forward processes. We will also consider starting Section 2 with a description of RHMC and its time reversal, using this simple process to illustrate the main ideas to the reader in a more gentle way before discussing PDMPs in full generality. We will work on improving other sections, for instance including better discussions of our assumptions and additional material on the implementation of our algorithms, emphasising the main differences with diffusion models.
>
>
>
> [1] Bierkens, J., Bouchard-Côté, A., Doucet, A., Duncan, A. B., Fearnhead, P., Lienart, T., Roberts, G., & Vollmer, S. J. (2018). Piecewise deterministic Markov processes for scalable Monte Carlo on restricted domains. Statistics & Probability Letters, 136, 148-154.
>
> [2] Bierkens, J., Grazzi, S., van der Meulen, F., & Schauer, M. (2023). Sticky PDMP samplers for sparse and local inference problems. Statistics and Computing, 33, 8.
>
> [3] Yang, J., Łatuszyński, K., & Roberts, G. O. (2024). Stereographic Markov Chain Monte Carlo. arXiv. https://arxiv.org/abs/2205.12112

---

### Official Review · Reviewer_a5JX · 2024-07-09

**Soundness:** 4
**Presentation:** 4
**Contribution:** 4
**Rating:** 7
**Confidence:** 5

**Summary:**

The paper explores generative models utilizing PDMPs, a type of stochastic process characterized by deterministic motion interspersed with random jumps at random times. These models offer an alternative approach to diffusion-based generative models, which have become very popular in the AI community in recent years. The authors focus on three specific PDMPs: the Zig-Zag process (ZZP), the Bouncy Particle Sampler (BPS), and Randomised Hamiltonian Monte Carlo (RHMC). The authors leverage the existing literature on the time reversals of Markov jump processes, characterizing the time reversal of any PDMP under appropriate conditions. They show that the time reversal of a PDMP is itself a PDMP with characteristics related to the original process. The authors also specifically outline the characteristics of the time-reversal processes for ZZP, BPS, and RHMC. The jump rates and kernels of these time reversals admit explicit expressions based on the conditional densities of the PDMPs before and after jumps.

The authors provide bounds on the total variation distance between the data distribution and the distribution of their generative models, considering errors from the approximation of the backward PDMP’s characteristics and its initialization from the forward process’s limiting distribution. Some initial but promising numerical simulations on simple toy distributions are presented, showcasing the potential of PDMP-based generative models. The results support further investigation into this class of models on more challenging data structures.

**Strengths:**

Originality: The paper introduces a class of generative models based on piecewise deterministic Markov processes (PDMPs). This is a very novel idea which departs from the widely used diffusion-based models. In my opinion, this is a fresh perspective on generative modelling and opens up new avenues for exploration and potential improvements in various application areas. The key idea behind diffusion-based models is the derivation of the reverse time and process, and following a similiar line of thinking, the authors of this paper characterize the time reversal of PDMPs, which is particularly original. While time reversal in diffusion processes is well-studied, applying these concepts to PDMPs and providing explicit expressions for their jump rates and kernels is a significant contribution.

Quality: The paper presents a comprehensive theoretical framework for PDMP-based generative models. The mathematical rigour in characterizing time reversals, deriving error bounds, and proposing training procedures leads to a complete and high-quality piece of work. The thorough analysis of the three specific PDMPs (ZZP, BPS, RHMC) and the detailed exploration of their time-reversal characteristics demonstrate a deep understanding and careful consideration of the underlying processes. The numerical simulations provide supporting empirical validation of the proposed models, adding credibility to the theoretical claims. The use of multiple toy distributions (extras are in the appendix) to test the models in the paper strengthens the evidence for their potential efficacy in practice.

Clarity: The paper is very well-structured, with a logical flow from the introduction of PDMPs to the detailed theoretical contributions and empirical results. Each section builds on the previous one, making it easier to follow the progression of ideas. The explanations of the PDMPs, their time reversals, and the derivation of jump rates and kernels are clear and very detailed. The inclusion of propositions and their proofs in the appendices provides a robust foundation for the proposed algorithms and establishes their convergence properties. The use of toy distributions to demonstrate the numerical simulations helps illustrate the practical application of the models, aiding in the reader’s understanding and also provides some nice opportunities for the authors to clearly articulate the advantages of their approach over the diffusion-based alternative approach.

Significance: The proposed PDMP-based generative models have the potential to be applied in a wide range of fields, from machine learning and statistics to physics and biology. This broad applicability enhances the significance of the work. By offering an alternative to diffusion-based models, this paper advances the field of generative modelling. The potential advantages of PDMPs, such as better scalability and reduced computational complexity, could lead to significant improvements in high-dimensional data generation. The theoretical insights and empirical results lay a strong foundation for future research in this area.

**Weaknesses:**

Originality: While the introduction of PDMP-based generative models is novel, the paper primarily focuses on theoretical aspects and toy datasets. There is a limited exploration of how these models can be applied to more complex, real-world problems, which could showcase their true originality and practical utility. Although PDMPs offer a new approach, the paper does not extensively compare these models with a wide variety of existing generative models, even on the simple toy examples that were considered here. This makes it difficult to fully appreciate the originality and benefits of PDMPs over other state-of-the-art methods.

Quality: The paper relies on several technical assumptions and conditions (e.g., H3, H4). While these are necessary for the theoretical results, their practical applicability might be limited and the authors do not discuss whether or not these assumptions hold for the toy examples which they consider. The paper could be strengthened by discussing the feasibility of these assumptions in real-world scenarios. The empirical validation is primarily limited to simple toy distributions. While these are useful for initial validation, the lack of experiments on more complex datasets (e.g., image or text data) reduces the overall impact and persuasiveness of the empirical results. It would be beneficial to include comparisons on more challenging benchmarks.

Clarity: The paper uses dense mathematical notation and detailed proofs, which may be challenging for readers who are not specialists in stochastic processes or PDMPs. This is going to be challenging for the authors as the high-level of technical detail provided in the paper does lead to a very robust paper. However, perhaps more intuitive explanations or visual aids could help make the content more accessible, if not in the main paper then in the appendix. The practical implementation details, particularly regarding the training procedures and simulation methods, are somewhat sparse (even though there are more details in the appendix). Providing a step-by-step guide or pseudocode could help practitioners better understand how to apply the proposed methods. Given the space constraints, this would have to be added to the appendix. This is covered in the case of splitting schemes, but could perhaps be modified to be more user-friendly to people new to this area of research.

Significance: The paper’s significance is somewhat limited by the focus on theoretical and synthetic examples. Without demonstrating the effectiveness of PDMP-based models on real-world data, it is challenging to gauge their practical significance and potential impact in applied settings. While the paper claims that PDMPs offer better scalability and reduced computational complexity, there is limited empirical evidence to support these claims. Benchmarking the computational performance against existing generative models would provide a clearer picture of the advantages and limitations in terms of scalability and efficiency.

**Questions:**

- I've outlined quite a few points in the weaknesses section that the authors may wish to address as questions.
- Could the authors include computational time with their simulations?
- Under H2, the authors state that the condition is satisfied when "the tails of $\mu_*$ are sufficiently light." Could this be clarified as to what "sufficiently light" means? I'm interpreting this as lighter than Gaussian.
- A small query, in Proposition 2, on Line 166, it states that $\mu_0^X \otimes \mu_0^V$ on $\mathbb{R}^2d$. But how does this work for ZZP when $V=[-1,+ 1]^d$ (sorry - I can't get the curly brackets to work)?

---

> ### Author Rebuttal · Authors · 2024-08-06
>
> Thank you for your thorough review and for taking the time to read and appreciate the full extent of our paper. We value your detailed feedback and have addressed your concerns below.
> * *Weaknesses (originality, significance)*
> 	* Our paper focuses on establishing theoretical foundations and validating our approach with toy datasets. We highlight that our main contribution in this paper was proposing this novel methodology together with the theory for this class of methods. We believe this was a significant contribution, as deemed by the reviewer.
> 	* In the numerical experiments, PDMPs show promising results, with faster convergence to the data distribution in terms of computational time (see the pdf attached to this rebuttal, Figure 1). We agree that demonstrating their performance on more complex, real-world problems would showcase practical utility more effectively, and this is an important future direction we wish to work on.
> 	* We expect PDMPs to be successful at modelling data distributions supported on restricted domains, as it is very simple to adapt their deterministic dynamics to incorporate boundary terms (see e.g. [1]), especially compared to diffusion processes. Similarly, data distributions that are a mixture of continuous density with point masses can be modelled adapting the ideas in [2], while extensions to data on Riemannian manifolds can be obtained leveraging flows which do not leave the manifold (see e.g. [3] for a PDMP on the sphere).
> 	* As the reviewer suggests, in the next revision we will work on comparisons with other generative methods (e.g., variational autoencoders and normalising flows) on the datasets we consider.
> * *Weaknesses (quality)* and *Questions: Under H2, the authors state that the condition...*
>
> 	We understand the remarks asking for more clarity regarding the assumptions that we require, and we agree that it is important to discuss whether these hold in practice. We will add these clarifications to the revised paper. We stress that H2 is the only assumption that depends on the data distribution. Let us comment on them.
> 	* H1 is about the chosen stationary distribution of the forward PDMP and is satisfied when this distribution is the multivariate standard normal, but also in the “variance exploding” variant, that is when $\psi(x) = 0$ for all $x$.
> 	* H2 is only required to prove Theorem 1 and is discussed in Appendix D.1. H2 holds when the stationary distribution of the PDMP is Gaussian and the data distribution has tails at least as light as those of a Gaussian in the case of ZZP and BPS, or in the case of HMC when it admits finite moments up to order 2. H2 is then satisfied in all our toy examples and in general when the data distribution is supported on a bounded set, as e.g. when the dataset is composed of images. We will give more information on this aspect in Appendix D.1 of the revised paper. We remark that in the case of ZZP and BPS much weaker conditions on tails of the data distribution can certainly be obtained by developing theory which is tailored for the context of generative modelling.
> 	* H3 is verified for the three PDMPs we consider, as mentioned in Appendix A.2. We will add comments on it in the appendix.
> 	* H4 is very technical, but is about the dynamics of the forward process rather than about the data distribution. It was verified in previous work for a specific version of BPS and can be verified for ZZP and RHMC after some technical arguments. Since this is out of the scope of our submission, we leave this verification for future works dedicated to this technical question.
> * *Clarity*
> We thank the reviewer for having found our paper theoretically solid. We agree that more intuitive and visual explanations could make the content more accessible. In order to improve the clarity of our paper, we will make the following changes.
> 	* Include a sample trajectory for the forward process of each sampler; you can find them in Figure 2 of the PDF attached to this rebuttal.
> 	* Add a dedicated ‘Training’ section in the appendix, which will be self-contained and include a step-by-step guide for each of the three PDMP considered. This will clarify the overall training procedures and make them easy to implement.
> 	* Another option we are considering is to start Section 2 with a description of RHMC and its time reversal, using this simple process to illustrate the main ideas and intuition in a more gentle and visual way. We could then outline the main ideas of the training procedures for the characteristics of RHMC, leaving the most technical details to a new appendix we plan to add. If the reviewer thinks this is a good idea, we would gladly make this addition to the revised version of our paper.
> * *Could the authors include computational time with their simulations?*
> As we previously mentioned, we include in the attached pdf a plot (Figure 1) displaying performance for each generative method on the Rose dataset, as a function of computational time. In addition, we plan to make the same type of comparisons on the other datasets that we consider, and add the resulting plots to the paper.
>
> * *A small query, in Proposition 2...*
> We consider that the uniform distribution on $\{ \pm 1 \}^d$ is also a distribution on $\mathbb{R}^d$, hence the proposition applies. Based on your comment we will add a clarifying remark.
>
> [1] Bierkens, J., Bouchard-Côté, A., Doucet, A., Duncan, A. B., Fearnhead, P., Lienart, T., Roberts, G., & Vollmer, S. J. (2018). Piecewise deterministic Markov processes for scalable Monte Carlo on restricted domains. Statistics & Probability Letters, 136, 148-154.
>
> [2] Bierkens, J., Grazzi, S., van der Meulen, F., & Schauer, M. (2023). Sticky PDMP samplers for sparse and local inference problems. Statistics and Computing, 33, 8.
>
> [3] Yang, J., Łatuszyński, K., & Roberts, G. O. (2024). Stereographic Markov Chain Monte Carlo. arXiv:2205.12112

---

> > ### Comment · Reviewer_a5JX · 2024-08-10
> > **Response to authors' rebuttal**
> >
> > Thank you to the authors for providing a comprehensive rebuttal. I am satisfied that the authors have addressed my questions and I thank them for the incorporating my feedback into their revised paper.

---

### Official Review · Reviewer_RL5x · 2024-07-16

**Soundness:** 3
**Presentation:** 2
**Contribution:** 4
**Rating:** 6
**Confidence:** 4

**Summary:**

This interesting paper on the popular topic of generative models introduce a new family of generative models which builds on the so-called piecewise deterministic Markov process (Zig-Zag process, Bouncy Particle Sampler, Randomised Hamiltonian Monte Carlo). In contrast to many of the existing models this family is not based on diffusion models. The paper includes a through analysis of the construction
and it propose training procedures and methods for approximate simulation of the reverse process.

**Strengths:**

* A new family of generative models is proposed.
* Thorough analysis of the properties of the proposed construction is provided.
* Simple examples provided.

**Weaknesses:**

* Missing real-world examples
* The is a big jump in  the style of writing between Section 1 and 2. Do not get me wrong here, the technical developments are most interesting, but many readers would be helped by a more gentle transition between these sections. Space for this can be created by moving more of the technical details into the supplemental material.

**Questions:**

* You provide convincing examples on simple synthetic datasets. Is you family of methods able to generate data also in more challenging real-world examples?
* When you compare the results of the various methods you show results for "steps". Is the a fair way of comparing the methods? Would it not be better to have some kind of computational cost on the x-axis? My reasoning here is that a "step" is not a well defined concept and it could mean very different things to different methods. In any case it would be really interesting to see the performance with the computational cost on the x-axis.

**Limitations:**

Real-world examples missing.

---

> ### Author Rebuttal · Authors · 2024-08-06
>
> We thank the reviewer for their time invested in our paper and their feedback. We believe we addressed all the raised issues below.
>
> * *There is a big jump in the style of writing between Section 1 and 2...*
>     * We agree the description of a time inhomogeneous PDMP is technical. In order to alleviate this difference in style between the two sections, in the next version of the paper we will add at the beginning of Section 2 the trace plots that can be found in the pdf attached to the rebuttal (Figure 2). These plots show the dynamics of the three PDMPs we discuss in the paper and hence help the reader develop a clear idea of the noising processes and their time reversal.
>     * In addition, one option that we are considering is to first present our methodology for the specific case of RHMC, discussing the forward process and its time reversal, before introducing PDMPs in full generality. RHMC is perhaps the simplest of the three processes and the characteristics of its time-reversal are relatively straightforward to present and can be intuitively interpreted. We could then briefly outline the main ideas of the training procedures to estimate these characteristics, leaving the most technical details to a new appendix. If the reviewer thinks this is a good idea, we would gladly make this addition to the revised version of our paper.
>     * Finally, in the supplement, we will add a section dedicated to the simulation of the forward PDMPs, with special emphasis to the case in which the limiting distribution is a multivariate standard normal. We think these additions will give the reader a more gentle, visual introduction to PDMPs and their time-reversals.
>
> * *You provide convincing examples on simple synthetic datasets. Is you family of methods able to generate data also in more challenging real-world examples?*
>
>     We are happy the reviewer found the experiments on synthetic datasets convincing.  The numerical experiments, albeit on simple datasets, indeed suggest that it is worth conducting more extensive studies and comparisons on real-world datasets. For the moment, this paper laid the theoretical foundations to this new class of methods. Obtaining further results on more complex datasets is an important future direction that we wish to work on.
>
> * *When you compare the results of the various methods you show results for "steps". Is this a fair way of comparing the methods? ...*
>
>     We agree with the reviewer, and have added to the pdf attached to the rebuttal a plot (see Figure 1) which reports the computational time on the x-axis and the performance on the y-axis, for the “rose” dataset. This plot clarifies the computational cost of our algorithms as compared to DDPM, and we indeed observe clear improvements. We plan to make the same type of comparisons on the other datasets that we consider and add the resulting plots to the paper.

---

### Official Review · Reviewer_LXdU · 2024-07-21

**Soundness:** 3
**Presentation:** 3
**Contribution:** 3
**Rating:** 7
**Confidence:** 3

**Summary:**

This paper considers the development of generative models based on piecewise deterministic Markov processes. The key idea proposed in the paper is to use piecewise deterministic Markov processes instead of diffusions as the "noising process" of the generative model. This relies on the fact that time reversals of PDMPs are themselves PDMPs. Three specific instances of PDMPs are considered. The authors also derive a bound (in total variation distance) between the data distribution and the distribution of the generative model. The methodology is illustrated with some simple experiments.

**Strengths:**

I enjoyed reading the paper and I like the idea of considering alternative noising processes in the context of generative models. The paper covers both theory and provides an example showing the viability of these methods. The examples are sufficient and certainly the area seems worthy of further investigation.

**Weaknesses:**

To me, the descriptions of approximating the process characteristics with normalizing flows are unclear. This part should be written more with more details, perhaps in the supplement, as this is core to being able to reproduce the results. I do appreciate that the authors provided a description of the experiment in E.1, but it is not enough to put things together. I would be interested in replicating at least the simple experiment, but I don't think I can do it as the paper stands.

**Questions:**

Are there particular data characteristics that these processes would be most suitable for? It seems that images with sharp boundary transitions are better modeled by the authors proposed approach. Would you say that is the case?

Do these methods have issues like being only able to generate from one mode if the data distribution is multimodal?

Why does RHMC do so well for such a small number of steps?

**Limitations:**

Yes.

---

> ### Author Rebuttal · Authors · 2024-08-06
>
> We thank the reviewer for their time invested in our paper and the relevant questions. Below, we believe we address all the raised concerns.
>
> * *To me, the descriptions of approximating the process characteristics with normalizing flows are unclear. ...*
>
>     We understand that we did not provide sufficient details on our methodology to obtain approximations of the characteristics of BPS and RHMC via normalising flows. We will clarify our implementation further both in the main text and in the supplement of the updated version of the paper, including pseudo-algorithms and step by step guides to reproduce the experiments.
>
>     * Let us give a brief overview of the methodology here. Our procedure to approximate the characteristics of the backward BPS and RHMC is built on minimising an empirical counterpart of the (maximum likelihood based) loss displayed between lines 252 and 253. Such empirical loss is obtained with the standard approach of using Monte Carlo estimates for the inner expectation as well as the integral with respect to the probability distribution on the time variable. Concerning the time variable, we draw a time $\tau$ independently from $\omega$ and use it as the time horizon for the forward PDMP, obtaining $(X_\tau,V_\tau)$.
>
>     * Then, the normalising flows framework is used to model the conditional density that appears in the loss. Essentially, we define a normalising flow that takes as input a $d$-dimensional standard normal random vector and outputs another $d$-dimensional random vector. The architecture of the normalising flow is described in Appendix E.1. Then, the normalising flow must be conditioned on the position and time variable since it models a conditional density. We achieve this by embedding the position and time via MLPs (with size given in Appendix E.1) and then injecting the outputs in the NF architecture at different points. The output of the NF is then a random vector which is approximately distributed according to the conditional distribution of v given x and t. The obtained conditional NF defines an invertible deterministic mapping $ v \mapsto T_{(x,t)}(v)$ and hence gives an explicit expression for the modelled conditional density by the change of variables formula. Therefore, this can be used to compute the empirical loss.
>
> * *Are there particular data characteristics that these processes would be most suitable for? It seems that images with sharp boundary transitions are better modeled by the authors proposed approach. Would you say that is the case?*
>
>     This is a very important question, which will be best answered through extensive experimentation. As the reviewer noticed, our numerical simulations indeed suggest that sharp boundary transitions are modelled well by our algorithms compared to DDPM. We also expect PDMPs to be successful at modelling data distributions that are supported on restricted domains, since it is very simple to adapt their deterministic dynamics to incorporate boundary terms (see e.g. [1]), especially if compared to diffusion processes. Similarly, data distributions that are a mixture of continuous density with point masses can be modelled adapting the ideas in [2], while extensions to data on Riemannian manifolds can be obtained leveraging flows which do not leave the manifold (see e.g. [3] for a PDMP on a sphere).
>
> * *Do these methods have issues like being only able to generate from one mode if the data distribution is multimodal?*
>
>      While the non-reversibility of PDMPs can lead to improved convergence, it is not yet clear whether this is beneficial in the context of multimodal distributions. Nonetheless, it is worth mentioning our 2D experiment with the Gaussian grid, which was designed to test the robustness of the different generative methods with respect to mode-collapse. The Gaussian grid at hand is an unbalanced multimodal dataset (see line 686), and we can see that the PDMP samplers all have better performance than DDPM with respect to the MMD metric. One can visually check on Figure 2 that the data coverage of RHMC compares favourably to DDPM. This suggests that PDMPs can generate from multimodal distributions.
>
> * *Why does RHMC do so well for such a small number of steps?*
>
>     This is an interesting and perhaps surprising behaviour. Our intuitive explanation is that velocity refreshments can guide the position vector of the process towards the data distribution quite efficiently assuming the last step is performed close enough to the time horizon and assuming the conditional distribution of the velocity vector given the position vector is learned accurately enough. This intuition is supported by the fact that this phenomenon is not observed for the Zig-Zag process when the refreshment rate is set to 0, in which case the model requires a larger number of reverse steps to give good performance. We will add a comment about this in the paper.
>
> [1] Bierkens, J., Bouchard-Côté, A., Doucet, A., Duncan, A. B., Fearnhead, P., Lienart, T., Roberts, G., & Vollmer, S. J. (2018). Piecewise deterministic Markov processes for scalable Monte Carlo on restricted domains. Statistics & Probability Letters, 136, 148-154.
>
> [2] Bierkens, J., Grazzi, S., van der Meulen, F., & Schauer, M. (2023). Sticky PDMP samplers for sparse and local inference problems. Statistics and Computing, 33, 8.
>
> [3] Yang, J., Łatuszyński, K., & Roberts, G. O. (2024). Stereographic Markov Chain Monte Carlo. arXiv. https://arxiv.org/abs/2205.12112

---

> > ### Comment · Reviewer_LXdU · 2024-08-08
> >
> > Thank you for these clarifications. I am satisfied with the responses.

---

### Official Review · Reviewer_FgP1 · 2024-07-25

**Soundness:** 3
**Presentation:** 2
**Contribution:** 4
**Rating:** 7
**Confidence:** 4

**Summary:**

The paper proposes using Piecewise Deterministic Markov Processes (PDMPs) for generative modelling applications, by using the property that PDMPs also admit time reversals that themselves are PDMPs. There are three major contributions in my understanding -

By characterizing certain families of PDMPs, i.e. Zig-Zag processes (ZZP, the Bouncy Particle Sampler (BPS), and the Randomised Hamiltonian Monte Carlo (RHMC), in terms of their jump rates and kernels, this paper shows how to obtain tractable closed-form and approximations for the jump rates and kernels of the time reversed PDMPs.
Theoretically, this paper then proposes a total variation bound between the “learnt” data distribution and the true data distribution when the base distribution is a Gaussian. This is a useful property, quite similar to bounds that have been proposed before in the literature (for example, for the Ornstein-Uhlenbeck process in [1, Theorem 5.2.]).
Finally, the paper proposes two empirical techniques to learn the time reversals akin to score-matching. First, for the ZZP process, inspired by score-matching techniques, the authors propose a ratio-matching technique. Secondly, for the BPS and RHMC processes, the authors learn normalising flows for the time reversal. They then show promising results in low-dimensional and MNIST generative modelling applications as a proof-of-concept.

[1] Dominique Bakry, Ivan Gentil, Michel Ledoux, et al. Analysis and geometry of Markov diffusion operators, volume 103. Springer, 2014.

**Strengths:**

I quite like the structure and formulation of the paper. The main goals and approach is elucidated quite clearly, and the mathematical preliminaries, while dense, seem correct for me.

In my understanding, it is a known fact that all PDMPs admit an equivalent time-reversal, but these are quite hard to calculate in general. In this paper, building on theory involving jump Markov Processes in [2], the authors derive expressions for time-reversal jumps and kernels for 3 different PDMPs. In general, deriving the backwards time reversal and then also designing an empirical scheme with a neural network architecture and loss function would be a substantial contribution, but this paper has many additional contributions on top of that. The numerical experiments seem compelling, even if a little small scale. However, this paper seems like a proof-of-concept on the use of time-reversed PDMPs for generative modelling, and I think the theoretical contributions along with the design of the loss functions and training paradigms are a pretty significant contribution already.

[2] Giovanni Conforti and Christian Léonard.   Time reversal of markov processes with jumps under a finite entropy condition. Stochastic Processes and their Applications, 144:85–124, 2022.

**Weaknesses:**

Fundamentally, I think the paper lacks a convincing argument about why generative modelling with PDMPs would fundamentally be more useful than traditional generative modelling. I understand that there were some arguments made in the introduction of the paper, namely Lines 35-36 (“such as better scalability and reduced computational complexity in high-dimensional settings”). However, it is really unclear to me how this argument actually translates to the empirical score-matching (or normalising flow training) objectives that the authors formulate, vs an approach like DDPM. The experimental section is quite lacking in details and comparisons about how the PDMP approach improves along any number of axes, beyond the qualitative plots. For example, I can think of many axes of improvement that could be discussed -
sample efficiency (how many training datapoints are needed to learn the time-reversal given that the process is partly deterministic),
mixing rates towards the Gaussian for their time reversal. Usually, SDEs such as the Ornstein-Uhlenbeck process are quite quick at mixing towards a Gaussian, making them quite nice to use when reversing a Gaussian distribution as the base. For partly deterministic processes, is this easier or harder to do?
Are there any comparisons to regular Markov process methods that can show that having an irreversible Markov process is beneficial here? I believe that this is a big factor in why PDMPs are alluring, and their irreversibility makes them mix faster and use less data [3]. Any experiments showing sample efficiency and mixing rates would be really beneficial here.

I am worried that there are many subtleties in the training and sampling procedures of diffusion models, and indeed there are many papers focusing solely on the empirical training tricks that can improve generative modelling, and comparing to a vanilla DDPM model doesn’t properly ablate the technique. I would be hesitant to rely on these empirical results as a surefire sign of improved modeling, which is frustrating, as theoretically, the paper does seem to point to this being the case, and I really do want to believe.

Furthermore, I think the paper could benefit from being a lot clearer about the specific advantages of PDMPs vs other stochastic processes for generative modeling. This is barely mentioned, but does form the crux of the empirical results. This made it difficult for me to read through the theoretical developments, proofs and theorems without knowing the reason why we would really want to do this in the first place.

I also think the paper can also benefit from being more explicit in how their developed score matching and normalizing flow training differs from traditional methods (maybe an algorithm block), as this would be something really interesting to practitioners looking to adopt existing codebases to using PDMPs instead.

[3] Bierkens, J., Fearnhead, P., and Roberts, G. (2019), “The zig-zag Process and Super-Efficient Sampling for Bayesian Analysis of Big Data,” The Annals of Statistics, 47, 1288–1320. DOI: 10.1214/18-AOS1715

**Questions:**

I’ve summarised a few of my questions in the Weaknesses section, but I list a few more questions here I would love some clarity on -

You mention that [2] originally suggested time reversal theorems for jump Markov processes. I found it really difficult to figure out what the exact distinctions are between the theoretical formulations of [2], and the specific novelty of this paper. I’d love clarifications on what the additional mathematical frameworks introduced are, that adopt this framework to PDMPs.
I would really like a better discussion on Equation 9, talking about what the different components of the inequality correspond to, and if there are any insights to gain about why PDMPs perform better or differently to something traditional like an OU process.
In Line 238-239, you mention that it’s easy to subsample the loss across dimensions. Is this still an unbiased, consistent estimator for the continuous time loss?
You mention that PDMPs typically have several types of jump rates and kernels, and provide Equation 2 as a parametric family of possible jumps and kernels. Is this the only possibility for constructing jumps and kernels?
At the end of each section describing one of the 3 PDMPs, the authors mention that there has been work showing exponential convergence to invariant distributions for all the PDMPs. How does this exponential convergence compare to the OU process? Is there actually faster convergence due to the inherent irreversibility of PDMPs?

**Limitations:**

Yes, they have addressed any potential limitations.

---

> ### Author Rebuttal · Authors · 2024-08-06
>
> We thank the reviewer for their valuable review. We hope the following responses will answer their main remarks.
>
> * We will include a paragraph in the introduction motivating the PDMP approach, based both on theoretical and empirical aspects. We expect PDMPs to be successful at modelling data distributions that are supported on restricted domains, since it is very simple to adapt their deterministic dynamics to incorporate boundary terms (see e.g. [1]), especially when compared to diffusions. Similarly, data distributions that are a mixture of a continuous density with point masses can be modelled adapting the ideas in [2], while extensions to data on Riemannian manifolds can be obtained taking advantage of flows which do not leave the manifold.
> * We acknowledge that the current focus of our paper has been on establishing the theoretical foundations and validating our approach with toy datasets. We underline that developing the theoretical foundations for this class of methods is non-trivial, which is our main contribution in this paper. The experiments on 2D data show faster convergence to the data distribution in terms of the number of reverse steps (Table 2 in the paper), which also translates in better computational time (see the PDF attached to the rebuttal, Figure 1). As the reviewer suggested, we will work on empirical studies to highlight the specific advantages of PDMPs over other methods, focusing in particular on sample efficiency. Regarding the mixing rates of PDMPs, improved convergence over traditional, reversible methods has been observed first in the physics literature [4], while quantitative estimates on the rate of convergence to the stationary distribution were obtained using the hypocoercivity approach in [5], hence these can be compared to existing results for diffusions for a given limiting distribution. Moreover, the recent work [6] shows that PDMPs can at most achieve a square root speed-up compared to the overdamped Langevin diffusion in terms of relaxation time, and that this can be achieved in some cases.
> * Contrary to what was erroneously written in the manuscript, we compared our model to the improved DDPM framework of [7], which already improves over the standard technique. While it is clear that this does not represent the current state of the art, we also found it to be a reasonable comparison, since our method is novel and, as the reviewer pointed out, there are many engineering tricks improving the performance of diffusion based models.
> * We agree with the reviewer that it is beneficial to give clearer explanations on the training phase of our models. We will provide a dedicated ‘Training’ section in the appendix of the revised paper, that will include algorithm blocks for each sampler and will highlight how our training procedures differ from well-established methods. We will also give more details on other aspects as for instance the simulation of the forward process.
>
> * *On the novelties compared to [8]* The work of [8] characterises the time reversals of a wide class of Markov processes with jumps. However, the conditions of [8] are abstract and stated in a “language” that differs from the standard in the context of PDMPs. Our contribution is twofold: (i) in Proposition 1 we give simple assumptions on the characteristics of the PDMP under which the process admits a time reversal, for which we give the backward characteristics; (ii) in Proposition 2 we consider the two abstract types of transition kernels that are used in the PDMP literature, namely deterministic mappings which change the velocity vector only and velocity refreshments, and give explicit formulas for the corresponding backward rates and kernels. Moreover, in Propositions 4,5, and 6 in the appendix we give clean statements for the time reversals of ZZP, BPS, RHMC, where the only assumption we require is a technical condition on the domain of the generator.  Therefore, while the abstract machinery for time reversals was developed in [8], in our paper we specialise it to PDMPs pruning the statement to the cleanest form possible.
>
> * We will provide a clearer explanation on the different terms in Equation (9) in the next version of the paper. The first term corresponds to the error caused by initialising the backward process at the limiting distribution instead of at the law of the forward process at time $T_f$. The second term is a consequence of using approximate backward rates defined with the estimated density ratios.
>
> * *On lines 238-239* If we draw uniformly at random a subset of the components $i=1,\dots,d$, then the expectation of the empirical loss coincides with the theoretical loss $\ell_I$ given in Proposition 3. Hence the estimator is unbiased, although its variance will increase.
> * In the classical framework of PDMPs, a PDMP is indeed constructed by specifying the rates and transition kernels for each type of jump.
> * The papers we cite at the end of the description of each PDMP on page 3 do not give quantitative upper bounds on the rate of convergence to the limiting distribution, hence they cannot be used to compare these PDMPs to e.g., the OU process. As mentioned above, there is both theoretical and empirical evidence that PDMPs can give faster convergence to a given probability distribution.
>
> [1] Bierkens, et al. (2018). Piecewise deterministic Markov processes for scalable Monte Carlo on restricted domains.
>
> [2] Bierkens et al. (2023). Sticky PDMP samplers for sparse and local inference problems.
>
> [4] Michel et al. (2015). Event-chain Monte Carlo for classical continuous spin models.
>
> [5] Andrieu et al. (2021). Hypocoercivity of piecewise deterministic Markov process-Monte Carlo.
>
> [6] Eberle et al. (2024). Non-reversible lifts of reversible diffusion processes and relaxation times.
>
> [7] Nichol et al. (2021). Improved Denoising Diffusion Probabilistic Models.
>
> [8] Conforti et al. (2022). Time reversal of Markov processes with jumps under a finite entropy condition.

---

> > ### Comment · Reviewer_FgP1 · 2024-08-14
> >
> > Thank you for your responses, they've definitely cleared up a few doubts I had regarding the paper. I think the paper is in a great spot with the addition of your rebuttal figures (1 and 2 in the pdf), and I am quite confident with my initial score, and will be happy to advocate for a paper acceptance, pending other reviewers' responses. Overall, I think the rebuttal responses were well-explained, and I thank the authors for the additional effort, and I'm confident this will help make the paper stronger.

---

### Author Rebuttal · Authors · 2024-08-06

We thank the reviewers for their constructive feedback, which will certainly help us improving the quality and exposition of our contribution. We have responded individually to each review to address their specific questions and remarks.

As part of our answer to some of the reviewers, we have attached a PDF document including two figures:
* *(Figure 1)* We provide a figure displaying performance on the Rose dataset as a function of *computational time*, for each generative method. We re-adapt the results listed in Table 2, where performance is given as a function of reverse steps. The figure illustrates that PDMPs improve over DDPM when the computational time is considered. We plan to make the same analysis on the other datasets that we consider and add the resulting plots to the next revision of the paper.
* *(Figure 2)* We provide sample trajectories for each PDMPs considered in order to visually illustrate their respective dynamics. These plots are part of our efforts to make the presentation in Section 2 more accessible for the reader, taking advantage of visual aids to introduce this class of generative models.


Please let us know if we have adequately addressed your comments and questions. We remain at your disposal for further clarifications. Thank you again for your engagement.

---

### Decision · Program_Chairs · 2024-09-25

**Decision:**

Accept (poster)

**Comment:**

The reviewers, some of whom are experts in topics intimately related to PDMPs such as MCMC methods, and others of whom are experts in generative modeling, unanimously recommend acceptance. I concur.